# High-throughput behavioral screen in *C. elegans* reveals Parkinson's disease drug candidates

Salman Sohrabi[1], Danielle E. Mor[1], Rachel Kaletsky [1], William Keyes[1] & Coleen T. Murphy [1✉]

We recently linked branched-chain amino acid transferase 1 (*BCAT1*) dysfunction with the movement disorder Parkinson's disease (PD), and found that RNAi-mediated knockdown of neuronal *bcat-1* in *C. elegans* causes abnormal spasm-like 'curling' behavior with age. Here we report the development of a machine learning-based workflow and its application to the discovery of potentially new therapeutics for PD. In addition to simplifying quantification and maintaining a low data overhead, our simple segment-train-quantify platform enables fully automated scoring of image stills upon training of a convolutional neural network. We have trained a highly reliable neural network for the detection and classification of worm postures in order to carry out high-throughput curling analysis without the need for user intervention or post-inspection. In a proof-of-concept screen of 50 FDA-approved drugs, enasidenib, ethosuximide, metformin, and nitisinone were identified as candidates for potential late-in-life intervention in PD. These findings point to the utility of our high-throughput platform for automated scoring of worm postures and in particular, the discovery of potential candidate treatments for PD.

[1] Department of Molecular Biology & Lewis Sigler Institute for Integrative Genomics, Princeton University, Princeton, NJ 08544, USA.
✉email: ctmurphy@princeton.edu

Parkinson's disease (PD) is a neurodegenerative movement disorder that is characterized by loss of dopaminergic neurons in the substantia nigra and formation of abnormal protein aggregates containing α-synuclein. Currently, there is no cure for PD, and treatment options only mitigate symptoms without modifying the course of the disease[1]. Using *diseaseQUEST*, our recently developed tissue-network approach to identify and test new candidates for human disease genes, we discovered a novel link between branched-chain amino acid transferase 1 (*BCAT1*) and PD[2]. We found that *BCAT1* expression is decreased in substantia nigra of PD patients, and reduction of *bcat-1* in *C. elegans* promotes neurodegeneration of cholinergic neurons and dopaminergic neurons expressing human α-synuclein. Moreover, adult-only RNAi-mediated knockdown of *bcat-1* in *C. elegans* causes age-dependent spasm-like 'curling' behavior, serving as a new model for PD motor symptoms[2].

The ability to perform large-scale screening for modifiers of PD-like curling dysfunction in *C. elegans* may uncover novel disease mechanisms and identify potential therapeutics. However, manual recording and quantification of curling in a liquid thrashing assay is labor-intensive and time-consuming[2]. Computer-aided protocols can considerably expedite the analysis of *C. elegans* behavioral phenotypes. Worm tracking software packages have been developed to quantify various aspects of locomotion such as velocity (forward and backward speed), curvature of the sinusoidal shape (bend angles and amplitude), and directional changes[3–6]. A package developed in the early 2000s successfully tracked *C. elegans* curling by checking when the binary image of a worm had a hole or the length of the centerline was <75% of the average[7]. More recently, a deformable shape estimation algorithm was utilized to model worm crawling motion, and was robust to coiling and entanglement[8]. Stable detection of self-occluding body shapes has also been achieved by exploiting low-dimensional worm postures for effectively capturing omega and delta turns[9].

Previous trackers require consecutive frames to resolve self-occluding postures, and as a result require the analysis of large amounts of image data from videos[10]. Although robust, these algorithms still require user inspection and supervision to correct errors[7–9]. Heavy data overhead and time-consuming manual inspection of track histories limit the throughput of these approaches. Single-frame analysis is therefore a superior alternative for some high-throughput assays[11–13], and can eliminate the propagation of errors by not relying on neighboring frames[14]. While one study has described a single frame-based algorithm that can detect curled worms[14], the image processing technique in this study requires high photograph quality to extract each worm's self-occluding outline. In addition to the high level of complexity of this method, the need for in-depth post-inspection to avoid errors and thresholding for quantification of behavioral phenotypes by the user limits throughput of such approaches.

The use of machine learning (ML) enables the user to go beyond subjective scoring for the development of more tolerant and more reliable algorithmic solutions[15,16]. For instance, WorMachine, a machine learning-based phenotypic analysis tool, allows scoring of continuous-sex phenotype and the quantification of fluorescent signals[17]. In the current study, we have created a ML-based algorithm, *C. elegans* **Sn**apshot **A**nalysis **P**latform (*Ce*SnAP), paired with a rapid snapshot acquisition technique that serves as a simple and versatile workflow to quantify *C. elegans* behavioral phenotypes. *Ce*SnAP can be used for smart analysis of videos or snapshots upon training of a convolutional neural network (C-NN). The program maintains low data overhead, eliminates the need for user supervision, and can be utilized by investigators with no computational background, greatly accelerating efforts to perform high-throughput screens. Using *Ce*SnAP, we performed high-throughput curling analysis of a total of 17,000 worms in order to identify drugs that ameliorate PD-like motor dysfunction, and found that enasidenib, ethosuximide, metformin, and nitisinone are promising candidates for repurposing to PD.

## Results and discussion

**Curling assay.** Similar to PD motor symptoms, spasm-like curling behavior of *bcat-1* RNAi-treated worms is both age-dependent and progressive[2], therefore we perform curling tests on day 8 of adulthood. To carry out a manual thrashing assay, 100 worms per treatment group are individually picked into M9 buffer, and 30 s videos of ~10 worms each are recorded (Supplementary Movie 1). A standard stopwatch is used to measure the percentage of time each individual worm spends in a curled pose over the span of each 30 s video[2]. As an example, to analyze five potential drug candidates and two relevant controls (Fig. 1a) using the manual curling assay, it would take a total of 15 h to capture videos and analyze the data (Fig. 1b). In contrast, using *Ce*SnAP to analyze the same number of videos reduces the time of analysis to 2–3 min, for a total time of ~2.5 h for video capture and analysis (six times faster than the manual assay; Fig. 1c).

Our high-throughput workflow uses snapshots instead of videos in order to maintain low data overhead and greatly simplify the quantification process. Worms from each treatment group are first washed with M9 buffer into 35 mm petri dishes secured on an orbital shaker. Pipetting worms from this continuously rocking setup into the wells of a 96-well plate ensures rapid, near-uniform distribution of worms across the wells (Supplementary Fig. 1a). A motorized microscope stage is programmed to automatically take one snapshot of each filled well until all wells are imaged, and to repeat this process $n$ times for a total of $n$ snapshots per well. Depending on the number of conditions and replicates, it would take 30 s to 2 min for the same well to be photographed again. As an example, the entire process of washing worms, filling 70 wells, and recording images for seven treatment groups requires only 20 min of benchwork (Fig. 1d). Importantly, *Ce*SnAP image analysis is completely autonomous and requires no user intervention after the neural network is trained. This high-throughput automated curling assay can record and analyze experimental data 40 times faster than the manual thrashing assay (Fig. 1b, d).

***Ce*SnAP.** The snapshot analysis platform is comprised of three modules, SnapSegment, SnapTrain, and SnapMachine, which enable users to carry out image segmentation, training of the neural network, and automatic quantification of snapshots, respectively (Fig. 1e). Using conventional thresholding techniques, SnapSegment extracts mask images of worms and can provide preliminary classification using dimensionless shape factors. In the next step, thousands of these cropped photographs can then be loaded onto the SnapTrain workspace to be manually categorized into different classes. SnapTrain can then train and optimize convolutional neural networks using this image database (Fig. 1f). Upon loading a desired neural network, SnapMachine can look for target subgroups in image stills and classify them (Supplementary Fig. 2a). Upon training a highly reliable neural network, *Ce*SnAP can eliminate user intervention and post-inspection in analyzing snapshots.

We have used this platform to automate curling analysis. A convolutional neural network, CurlNet, has been trained on a total of 32,000 extracted mask images (Fig. 1f). In all, 16,000 images in this augmented database were manually labeled "Censored" due to entangled (non-interpretable) worms, debris, progeny, or highly noisy background (Fig. 1g). The remaining 16,000 worm images used for training were comprised of an equal number of Coiled, Curled, Near-curled, and Non-curled worms to provide equal chance of classification for each Worm

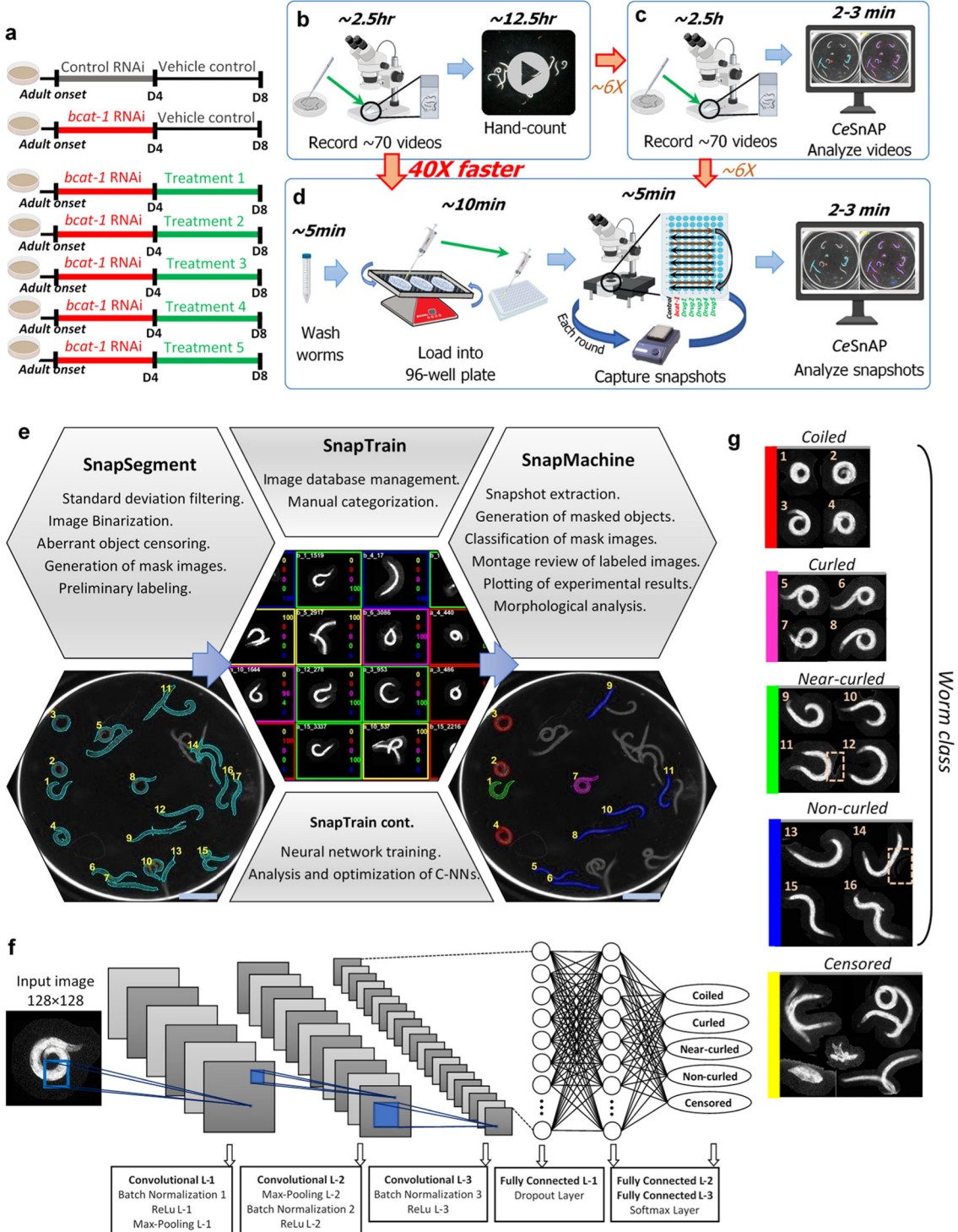

**Fig. 1 High-throughput automated screening platform for analysis of Parkinson's-like curling motor dysfunction in *C. elegans*. a** Experimental design for drug treatments. Neuronal RNAi-sensitive worms were fed *bcat-1* or control RNAi as adults, and on day 4 were transferred to plates with 50-μM drug or vehicle. Curling was measured on day 8. **b** The manual thrashing assay for five drug candidates and two relevant controls requires 15 hours of video recording and hand-counting analysis. **c** Using *Ce*SnAP to analyze videos is six times faster than the manual thrashing assay. **d** The high-throughput automated curling assay uses snapshots instead of videos and can be carried out with only 23 min of benchwork for the same or more test conditions. This automated workflow is 40 times faster than the manual assay. The time denoted for each operation indicates the time the user spends on each step, not including duration of automated image acquisition by the motorized stage or autonomous computational time. **e** *Ce*SnAP simple workflow (segment, train, and quantify) allows for fast analysis of snapshot-based assays after training a reliable neural network. Scale bar, 1 mm. **f** SnapTrain architecture. The convolutional neural network (C-NN) receives an input mask image of size 128 × 128 pixels and outputs scores for each category. **g** Sample mask images of Coiled (red), Curled (magenta), Near-curled (green), Non-curled (blue), and Censored (yellow) target classes used to train CurlNet. Dashed boxes depict deletion of neighboring entities, so the mask image only includes one 8-connected object before training.

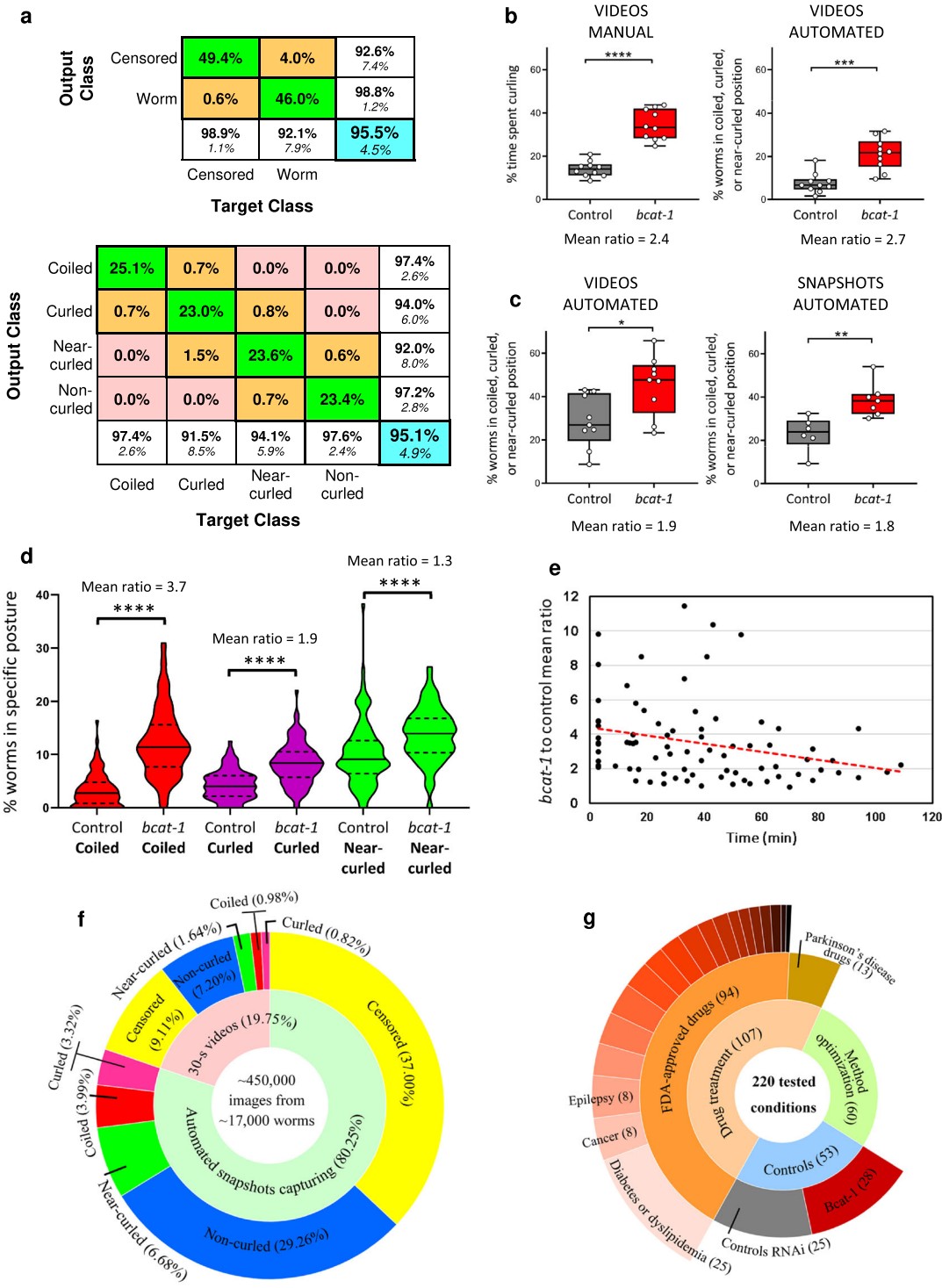

categories. Our trained network has 95.5% accuracy in correctly differentiating Worm versus Censored target classes in the test set (Fig. 2a). It should be noted that only 1.2% of Censored mask images were mistakenly classified as Worms. Similarly, CurlNet has a 95.1% success rate at correctly assigning Coiled, Curled, Near-curled, and Non-curled labels to our test database of Worm classes (Fig. 2a). The 4.9% classification error in differentiating different postures is largely due to transitional postures between two morphologically adjacent categories.

After training a reliable C-NN, SnapMachine is then able to automatically quantify snapshots of future experiments with high

accuracy. Upon loading CurlNet, SnapMachine extracts mask images of objects and classifies them into Coiled, Curled, Near-curled, and Non-curled categories (Supplementary Fig. 2a). Analyzing a test set of snapshots, we found that CurlNet censored 37.0% and 9.11% of worms during detection from snapshots and videos, respectively, as they were entangled or non-interpretable (Fig. 2f). Detection success rate decreased as the number of worms in each well increased due to the higher chance of overlapping worms (Supplementary Fig. 1b). CurlNet also detected Worm classes with reliable performance in front of a less clean background (Supplementary Figs. 2b, c and 3). Slightly distorted

**Fig. 2 Validation and performance of trained neural network for automated curling analysis. a** Confusion matrix plots of CurlNet. Rows represent the predicted class while columns correspond to the original class. The percentage of observations in the test set which were incorrectly classified are shown in off-diagonal cells. False discovery rates can be found in the column on the far right. Similarly, false negative rates are shown in the row at the bottom of the plots. CurlNet successfully distinguished Worm class with 98.8% accuracy (top). Objects in Worm class are classified into correct categories with 95.1% accuracy (bottom). **b** Comparison of manual hand-counting (left) versus software analysis (right) for the same set of 30 s videos of vehicle-treated worms showed significant curling detected by both methods. The automated analysis uses Coiled + Curled + Near-curled values, in order to best approximate the manual scoring criteria. n = 10 videos totaling 99 worms for control, 10 videos totaling 98 worms for *bcat-1*. **c** Comparison of automated analysis of 30 s videos (left) versus snapshots (right) (captured on the same day from two separate aliquots of worms and analyzed by including all three curling categories) showed that snapshots are sufficient to detect a significant difference between *bcat-1(RNAi)* and control. n = 5 wells totaling 50 worms for control videos, 7 wells totaling 73 worms for *bcat-1* videos, 6 wells totaling 71 worms for control snapshots, and 6 wells totaling 41 worms for *bcat-1* snapshots. **d** Meta-analysis of curling categories across 14 experiments showed Coiled posture distinguish spasm-like 'curling' motor defect of vehicle-treated *bcat-1(RNAi)* worms with greater mean ratios of *bcat-1* to control than that of Curled and Near-curled. n = 94 wells totaling 780 worms for control, 101 wells totaling 1160 worms for *bcat-1*. **e** The difference between control and *bcat-1* in terms of percentage worms in Coiled postures diminishes with time after transferring animals into liquid buffer. *bcat-1* to control mean ratio of each data point is calculated by averaging curling levels of replicates in one round of snapshots initiated at the reported time. Red dashed line represents a linear curve fit for rounds data points at different times. n = 89 rounds of snapshots across 17 experiments totaling 3166 worms. **f** Sunburst plot summarizing program performance in detecting and categorizing 17,000 worms. **g** Summary of experimental categories for all 220 tested conditions with number of conditions in parentheses. Two-tailed t-tests. *p < 0.05, **p < 0.01, ***p < 0.001, ****p < 0.0001. Violin plot represents probability density of percentage worms in specified posture. Box and violin plots show minimum, 25th percentile, median, 75th percentile, maximum. Mean of *bcat-1(RNAi)* divided by mean of control RNAi is abbreviated as mean ratio.

images due to non-uniform meniscus, out of focus photographs, bubbles, bacterial chunks, and debris are some example situations that may happen. We found that we only needed to manually override CurlNet decisions to censor or change posture labels for <2% of detected worms. The reason for this high accuracy is low classification error for mistakenly labeling censored mask images as Worm class (1.2%) and the fact that actual categorization error differentiating mask images of Worm class are <4.9%. With this negligible margin of error, SnapMachine is capable of analyzing any number of snapshots with virtually no dependency on the user's input. The *Ce*SnAP source code together with trained CurlNet and instruction manual are available on https://github.com/murphylab-Princeton/CeSnAP.

**Testing the high-throughput curling workflow**. To test the utility of our automated software and the use of snapshots instead of videos, we conducted several tests using vehicle (0.5% DMSO)-treated *bcat-1(RNAi)* and control RNAi-treated worms (Fig. 1a). We first tested the ability of *Ce*SnAP to replicate findings from the manual assay. The same set of 30 s videos were manually hand-counted and analyzed using *Ce*SnAP, which extracted 30 stills at 1 s intervals from each video (Fig. 2b). In the manual assay, the timer is started as soon as an animal starts to curl, which necessarily includes time when the animal is near-curled, prior to being fully coiled. Therefore, in order to best compare manual and automated analyses, worms labeled as Coiled, Curled, and Near-curled were summed by SnapMachine. The automated analysis was able to detect significantly higher curling in *bcat-1(RNAi)* worms compared to control, as did manual hand-counting. The ratios of *bcat-1(RNAi)* to control curling levels were similar, 2.4 in the manual assay compared to 2.7 in the automated quantification (Fig. 2b).

Next, we compared the use of snapshots versus videos in our automated curling analysis. We used a set of 30 s videos and snapshots that were captured on the same day from two separate aliquots of the same source of worms. Including all three curling categories in our analysis (Coiled, Curled, and Near-curled), we found that the mean curling level of *bcat-1(RNAi)* was 1.9-fold higher than that of controls using 30 s videos, and 1.8-fold higher using snapshots (Fig. 2c). The probability density of the percentage worms in Coiled, Curled, and Near-Curled postures across 14 experiments revealed that, while each curling posture is significantly higher in *bcat-1(RNAi)* worms compared to controls,

the Coiled posture presents the highest mean ratio of *bcat-1(RNAi)* to control curling level (Fig. 2d).

Re-analyzing videos-snapshot comparison test using only the Coiled class, the mean curling level of *bcat-1(RNAi)* was 12-fold higher than that of control using 30 s videos, and 3.4-fold higher using snapshots of first three rounds, though the difference between *bcat-1(RNAi)* and control was statistically significant in both cases (Supplementary Fig. 1c). We have found that episodic swimming of animals is responsible for this loss of sensitivity in snapshots. Worms are highly active in the first few minutes after transfer into liquid buffer[18]. Shortly thereafter, episodic swimming begins as worms enter intermittent inactive states for short periods of time[19]. Consistent with this observation, we found that the ratio of *bcat-1(RNAi)* to control curling levels gradually decreased during later rounds of snapshots (Fig. 2e). However, the analysis of snapshots (captured within a 60 min window after transfer to liquid) was sufficient to detect a highly significant difference between *bcat-1(RNAi)* and control, demonstrating the suitability of replacing 30 s videos with snapshots for high-throughput screening (Supplementary Fig. 1c).

It takes our linear stage ~2 s to photograph and move to the next location. During a 60-min time period, the motorized platform can capture up to 1800 snapshots. The user can distribute this total based on the number of conditions, replicates, and rounds in their experiments. Each round can last up to 15 min before animals start to enter an inactive state[18] and need to be re-stimulated on a thermoshaker. A total of 20 snapshots per well (three rounds of snapshots) was sufficient for each condition's curling analysis (Supplementary Fig. 1c). Users can capture as many snapshots as they see fit, since this will not increase the 25-min benchwork or memory load (Fig. 1d). We suggest three rounds of snapshots (seven snapshots per round per well) for no more than seven experimental conditions to ensure that each round finishes in under 15 min.

Using our high-throughput automated workflow, we were able to test more than 17,000 worms, with an average of 77 animals per condition and 12 worms per well. With a 53.89% detection rate for the Worm class, ~450,000 (17,000(worms) × 7(stills) × ~7(rounds) × 53.89%) mask images of individual worms were classified (Fig. 2f). Because of the negligible error rate, no manual censoring or label changes were deemed necessary. In total, 220 conditions were tested to optimize the curling assay and screen for drugs that improve motor function in *bcat-1(RNAi)* worms (Fig. 2g).

**High-throughput drug screen to identify potential PD therapies.** Neuronal RNAi-sensitive worms were fed *bcat-1* or control RNAi at the onset of adulthood until day 4, and were then switched to heat-killed OP50 *E. coli* bacteria with 50 µM drug or vehicle (0.5% DMSO). Curling assays were performed on day 8 (Fig. 1a). We specifically chose a mid-stage intervention (day 4) in order to mimic the PD patient who enters the clinic seeking treatment at a time when the disease is already underway. We first treated the worms with the motor-specific PD medications, selegiline, and trihexyphenidyl. Similar to PD patients who experience symptomatic relief upon administration of these drugs, curling was significantly reduced in *bcat-1(RNAi)* worms following treatment with these PD medications (Fig. 3a). These results support the use of the curling phenotype to model PD motor symptoms, and suggest that this experimental workflow can be used to identify potential therapeutics.

We next conducted a proof-of-concept screen of 50 FDA-approved drugs to identify those that improve *bcat-1*-related motor dysfunction in *C. elegans*. The 50 drugs in the screen were selected to represent a wide range of clinical indications (Supplementary Fig. 4a) and targets/mechanisms of action (Supplementary Fig. 4b). Half of the drugs are known to target

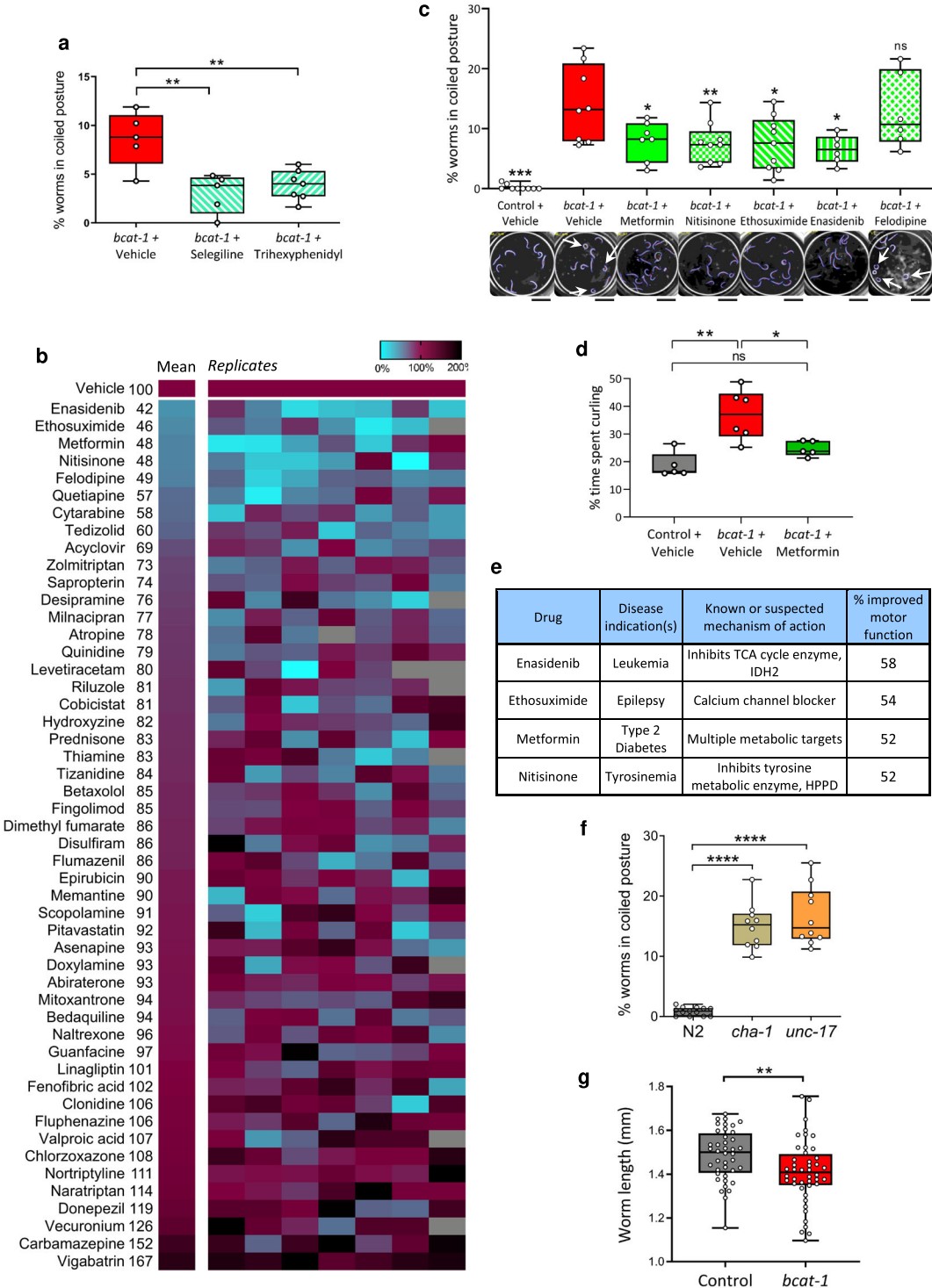

**Fig. 3 Application of high-throughput curling platform to the discovery of potential therapeutics for PD. a** Current Parkinson's therapies, Selegiline (monoamine oxidase inhibitor, an enhancer of dopaminergic signaling), and Trihexyphenidyl (anticholinergic), administered on *bcat-1* RNAi-fed worms on day 4 of adulthood improved motor function by 50% and 42%, respectively. $n = 5$ wells totaling 52 worms for *bcat-1*, 5 wells totaling 47 worms for Selegiline, 7 wells totaling 89 worms for Trihexyphenidyl. One-way ANOVA with Dunnett's post-hoc. **b** High-throughput screen of 50 FDA-approved drugs for those that improve motor function in *bcat-1(RNAi)* worms. Vehicle-treated level of *bcat-1(RNAi)* worm curling was set to 100%, and color indicates percentage worms in Coiled postures relative to vehicle. Blank wells are those that had <3 worms and were excluded. **c** Post-screen validation of top five drug candidates, which reduced curling to <50% of vehicle-treated levels. Software analysis of 30 s videos confirmed that 4 of the top 5 drugs reliably reduce curling. Values were normalized to vehicle-treated *bcat-1(RNAi)* from the same experiment, and statistics are shown for each drug group compared with *bcat-1(RNAi)* + vehicle. Scale bars, 2 mm. $n = 5$ videos totaling 52 worms for control RNAi, 7 videos totaling 74 worms for *bcat-1(RNAi)* + vehicle, 5 videos totaling 95 worms for metformin, 6 videos totaling 66 worms for enasidenib, 5 videos totaling 83 worms for nitisinone, 5 videos totaling 88 worms for ethosuximide, and 6 videos totaling 60 worms for felodipine. One-way ANOVA with Dunnett's post-hoc. **d** Manual hand-counting served as a final confirmation that one of the top-performing drug candidates, metformin, significantly improves motor function in the *bcat-1(RNAi)* worm model of PD. $n = 50$ worms for control vehicle, 57 worms for *bcat-1* vehicle, and 91 worms for *bcat-1* metformin. One-way ANOVA with Dunnett's post-hoc. **e** Disease indications and targets of the top 4 drugs that were confirmed to reduce curling. Percentage improved motor function is based on the results from the high-throughput screen. **f** Curling was detected in cholinergic mutants *unc-17* and *cha-1* on day 8 indicating that the high-throughput curling assay can be applied beyond the *bcat-1(RNAi)* model to other genetic backgrounds that induce curling. $n = 52$ worms for N2, 43 worms for *cha-1*, and 57 worms for *unc-17*. **g** Automated morphological analysis using continuous curvature values along the spine of Near-curled and Non-curled animals revealed *bcat-1(RNAi)* worms are significantly smaller than control worms. $n = 42$ non-curled or near-curled worm images for control, 44 non-curled or near-curled worm images for *bcat-1*. Two-tailed *t*-tests. ns not significant. *$p < 0.05$, **$p < 0.01$, ***$p < 0.001$, ****$p < 0.0001$. Box plots show minimum, 25th percentile, median, 75th percentile, maximum.

neurotransmission, including medications for epilepsy, depression and other mood disorders, schizophrenia, and the neurodegenerative diseases Alzheimer's and amyotrophic lateral sclerosis. Drugs with non-neurological indications were included to allow for the discovery of potentially novel therapies; current indications for these drugs include cancers, diabetes, infectious diseases, autoimmune disorders, and cardiovascular diseases.

Several drugs emerged from the screen as potential candidates for repurposing to PD (Fig. 3b). These drugs have not been used for PD before. Enasidenib, ethosuximide, metformin, nitisinone, and felodipine reduced curling to <50% of vehicle-treated levels (Fig. 3b). Including Coiled and Curled postures in curling analysis also showed reduction to <65% of vehicle-treated levels upon treatments with these hit candidates (Supplementary Fig. 4c). Post-screen validation using automated analysis of 30 s videos confirmed that 4 out of 5 of these drugs reliably and significantly reduced curling (Fig. 3c), and final confirmation for one of the hit candidates, metformin, was obtained by manual hand-counting (Fig. 3d). Detailed cellular and molecular characterization of metformin's neuroprotective action in *bcat-1(RNAi)* worms is reported in our companion study[20].

Variability across replicates (Fig. 3b) is an inherent part of the curling assay. We have generally used 7 or more wells, additional rounds of snapshots (Supplementary Fig. 1c), and follow-up reconfirmation videos (Fig. 3c) to mitigate the effect of this noise in identifying hit candidates.

To test if the top drug candidates might decrease curling for the trivial reason of disrupting the RNAi machinery, we deliberately knocked down a component that is necessary for RNAi processing, *dcr-1*. Moving *bcat-1(RNAi)* worms to *dcr-1* RNAi on day 5 did not rescue motor function on day 8, indicating that disrupting the RNAi machinery is not sufficient to restore motor function (Supplementary Fig. 4d). We also performed post-screen validation of three negative hits, riluzole, linagliptin, and milnacipran, and the results were in agreement with our original screen, suggesting that false negatives were unlikely in our screen (Supplementary Fig. 4e).

While only one of the four validated drugs from our screen, ethosuximide, is currently prescribed for a neurological disease, the known targets/activities of the four identified candidates appear to converge on similar cellular pathways related to metabolism (Fig. 3e). Metformin and nitisinone act on metabolic targets[21,22], enasidenib inhibits the TCA cycle enzyme IDH-2[23],

and ethosuximide is a calcium channel blocker[24]. These data suggest that drugs with metabolic and/or mitochondrial targets may represent a potentially new class of therapies for PD.

## Conclusion

*C. elegans* is highly amenable to high-throughput screening approaches, allowing rapid testing of potential disease treatment options[25]. Combined with an automated snapshot-capturing platform, our machine learning-based program can be trained to screen thousands of worms for phenotypic changes in a time-efficient manner. The simple segment-train-quantify workflow in *Ce*SnAP can be directly used or its open-source code can be modified for other snapshot-based assays (Fig. 1e).

To quantify spasm-like 'curling' behavior of *C. elegans*, our high-throughput curling platform uses a motorized stage for capturing a series of snapshots, and then CurlNet, a previously trained convolutional neural network, is loaded onto *Ce*SnAP workspace to carry out fast curling analysis. Worm detection and classification of postures is autonomous, unbiased, and requires no post-inspection, which makes it distinct from currently available tools. However, the lack of ability to track individual worms over time is the main limitation of our platform. For instance, *Ce*SnAP is unable to quantify locomotion measures such as speed or thrashing frequency. Furthermore, although using sparsely sampled stills is faster for screening, subtle behavioral phenotypes that require high frame-rate analysis may only be detectable using CeSnAP with low-throughput video-based assays.

We screened more than 17,000 worms at a rate 40 times faster than the alternative manual assay, testing for potential therapeutic compounds that may be repurposed for PD. We identified the FDA-approved drugs enasidenib, ethosuximide, metformin, and nitisinone as highly promising candidates for further investigation as potential late-in-life interventions in PD. The application of our high-throughput curling assay is not limited to worms with *bcat-1* knockdown; it can also be used to study Coiled, Curled, Near-curled, and Non-curled swimming postures with other RNAi treatments or in other genetic strains (such as cholinergic mutants *cha-1* and *unc-17*; see Fig. 3f). Additionally, a geometrical reconstruction of worm body in curvilinear coordinates (Supplementary Fig. 1d) enables high-throughput morphological analysis that does not require post-inspection (Fig. 3g). Collectively, these findings point to the utility of our high-throughput

platform for uncovering disease mechanisms and potential therapeutics.

## Methods

**C. elegans strains and maintenance.** C. elegans strains were grown at 20 °C on nematode growth medium (NGM) plates or high growth medium (HG) plates seeded with OP50 Escherichia coli or HT115 RNAi Escherichia coli. RNAi clones were obtained from the Ahringer RNAi library. The following strains were used in this study: wild-type worms of the N2 Bristol strain, CF512 (fem-1(hc17); fer-15 (b26)), TU3311 uIs60 (unc-119p::sid-1, unc-119p::yfp), CB113 (unc-17(e113)), and TY1652 (cha-1(y226)).

**RNAi and drug treatments.** For RNAi experiments, worms were synchronized from eggs by bleaching and placed on HG plates seeded with OP50. For CF512 (fem-1(hc17); fer-15(b26)), animals were sterilized by incubation at 25 °C from L2-L4. Sterilization of worms with FUdR was found to not be suitable for quantification of the curling phenotype. RNAi-seeded 100-mm NGM plates containing carbenicillin and IPTG were pre-induced with 0.1 M IPTG 1 h prior to transfer of worms at the L4 stage. In all experiments, control RNAi refers to empty vector pL4440 in HT115 Escherichia coli. Day 4 worms were transferred onto fresh NGM plates seeded with 1 mL heat-killed OP50 bacteria and 50-μM drug or vehicle (0.5% DMSO). OP50 was killed by incubation at 65 °C for 30 min. Curling was measured on day 8. Drugs for high-throughput screening were selected from the FDA-approved Drug Library (MedChem Express).

**Manual curling assay.** Approximately 100 worms per experimental condition were deposited 10–15 at a time into a 10 μL drop of M9 buffer on a microscope slide. Approximately 8–10 30 s videos were captured for each tested condition using an ocular-fitted iPhone camera attached to a standard dissection microscope and curling was manually quantified with a standard EXTECH Instruments stopwatch. Percentage time spent curling is defined as the sum of the periods in which either the head or tail makes contact with a noncontiguous segment of the body, divided by the total time measured[2].

**Automated image capturing.** On the day of analysis, animals were washed twice with M9 buffer and dispensed into 35 mm petri dishes rocking on an orbital shaker to ensure even distribution. Worms were pipetted with large orifice tips into the wells of a 96-well plate. In total, 30 s videos or a series of snapshots were obtained for individual wells containing up to 30 worms each. It is imperative to pre-fill 35 mm petri dishes and the 96-well plate with 6 mg/ml OP50 solution in M9 to prevent both starvation and worms sticking to surfaces. Before initiating each round of automated image capturing, worms in the 96-well plate were stimulated on a thermoshaker (Eppendorf) at 900 rpm for 60 s. Images were captured with inverted colors, i.e., white worms on a black background. To ensure efficient image capturing, rows were sequentially imaged in alternating directions. Shaking between each round is essential for restarting movement of worms in a state of quiescence and for creating a uniform meniscus to avoid distorted photographs.

**Statistics and reproducibility.** For all comparisons between two groups, an unpaired Student's t test was performed. For comparisons between multiple groups, One-Way ANOVA was performed with post-hoc testing. GraphPad Prism was used for all statistical analyses.

**CeSnAP.** SnapSegment can be used to extract mask images of objects from stills to be imported to SnapNet workspace. The purpose of SnapSegment and SnapNet toolboxes is to deliver a reliable trained neural network for use in SnapMachine to quantify snapshots of experiments. See CeSnAP instruction manual for more information which is available in program package.

**SnapSegment.** After importing raw images, the user roughly determines the circular outline of one well in the first snapshot by identifying three points on the well's circumference. To locate the animals in a single image, the gradient of image delineates the edges of the animals. Detection parameters such as min/max area-limit, Mask area, Levels-Otsu's method, N-node, and Jaggedness can be adjusted based on imaging setup or user preference. Detected objects are initially filtered through a minimum and maximum target surface area. The defaults are 0.1 and 8 times the average size of an adult worm. Mask area determines the size of the mask around found objects. Otsu's levels can accept combination values such as [1,1], [2,1], [2,2], etc., where the first number identifies how many multilevel thresholds will be calculated for the gradient image using Otsu's method, and the second number dictates which threshold will be used for initial binarization. The default values are set to [1,1]. N-node indicates the number of nodes used to outline each detected object with default value of 60. Jaggedness threshold automatically filters out found objects that do not have smooth perimeter. Jaggedness is defined as the perimeter of the originally found object to the perimeter of the smoothened one; the default value is 1.3.

In the Analyze subsection, the user can generate preliminary labels for masked images to facilitate manual categorization in SnapTrain workspace. Otsu's method is used to identify area and circumference outliers. The first number in Area filter and Perimeter filter indicates the number of multilevel thresholds used. Objects that have smaller or larger area/perimeter are labeled "Censored". Zero denotes no high or low band filtering. Additionally, Dim filter removes extremely thin or long objects. Utilizing dimensionless shape factors such as circularity and convex hull, SnapSegment can also categorize animals' postures into subgroups. CeSnAP source code can also be modified to add more thresholding criteria for preliminary labeling.

Mask images are either padded or rescaled to the network's defined input size of $128 \times 128$ pixels. They were also carefully masked to contain only one 8-connected object. The masks are dilated from a binarized image with a disk-shaped structuring element of radius ($4 \times A_{\text{object}}/P_{\text{object}}$). The labeling at this stage is preliminary and optional, as the SnapTrain module provides an interactive interface for manual label management.

**SnapTrain.** After loading mask image database onto a color-coded workspace, users can create up to 8 target classes and change object's labels with right/left clicks in this interactive montage. A convolutional neural network (C-NN) can be trained on augmented version of this image database. In C-NN architecture, convolutional layer 1 has 16 $3 \times 3$ convolutions with stride 1 and padding 2. The Max-Pooling layer has $3 \times 3$ max pooling with stride 1 and padding. Convolutional layer 2 has 32 $3 \times 3$ convolutions with stride 1 and padding 1. Max-pooling layer has $2 \times 2$ max pooling with stride 2 and padding 0. Convolutional layer 3 has 16 $3 \times 3$ convolutions with stride 1 and padding 2. Convolutional layer 3 has 64 $3 \times 3$ convolutions with stride 1 and padding 1. Fully connected layers 1, 2, and 3 have output sizes of 100, 100, and 5, respectively. Dropout layer's Probability is 0.25. The performance of trained C-NN can then be assessed and optimized as prediction scores are respectively shown in montage screen of SnapTrain.

To automate curling analysis, CurlNet has been trained on a total of 32,000 mask images divided between Censored and Worm classes. Equal numbers of Coiled, Curled, Near-curled, and Non-curled images in Worm class (totaling 16,000) were used to provide equal chance of detection for each worm categories. Entangled worms, debris, progeny, and non-interpretable mask images are labeled as Censored. Since trained network will be solely responsible for detection and classification, we meticulously censored worms with even minor faulty features. Our strict labeling approach ensured high reliability and eliminated the need for time-consuming post-inspection.

An augmented image database with random rotation and x/y reflections was split into 75% training set, 10% validation set, and 15% test set. Training was performed with a mini-batch size of 256, 0.9 momentum, L2 regularization of 0.0001, and a learning rate of 0.005 as images were shuffled and randomly rotated every epoch. Our neural network yielded 93.2% training accuracy, 92.3% validation accuracy, and 93.1% test accuracy. The trained neural network can assign five probability score to each mask image.

**SnapMachine.** After loading videos or snapshots on the SnapMachine workspace, the user can choose any trained network to carry out a fully-automated classification. Standard deviation filtered images of each snapshots are binarized using three multilevel image thresholds. 8-connected components from these three binarized images of the same still are then extracted. These binary components were dilated to generate four sizes of binary masks for each greyscale object. Scoring multiple mask images of same object by the trained neural network ensures elimination of the biased motive of thresholding technique used in extracting objects. Final decisions are made based on average scores of multiple mask images of same object. We found this methodology to be highly reliable (with >98% accuracy), which ultimately eliminates the need for post-manual inspections. The automated decision can still be overridden in the interactive montage page. Finally, the results can be exported for further statistical analysis or may be plotted within CeSnAP.

**Mathematical modeling of worm body.** A geometric model of the worm body has also been incorporated into CeSnAP framework for further morphological analysis. After applying a thinning algorithm to extract the centerline, worm anatomy is mathematically rebuilt in curvilinear coordinates using a two-dimensional surface around the centerline, where $u$ is the distance along centerline curve and $R(u)$ is the radial distance from the centerline[8]. The bending angle $\Theta(u)$ along the centerline is modeled using a third order B-spline composed of $N = 8$ splines. Similarly, the worm width profile $R(u)$ is defined as a continuous function of distance along its length (Supplementary Fig. 1d).

$$\Theta(u) = \sum_{j=1}^{N} \alpha_j \emptyset_j^3(u) \qquad (1)$$

$$R(u) = \sum_{j=1}^{N} r_j \emptyset_j^3(u) \qquad (2)$$

Cartesian coordinates of worm body can be expressed as

$$X(u, R(u)) = \int_0^u \begin{bmatrix} \cos\Theta(s) \\ \sin\Theta(s) \end{bmatrix} ds + R(u) \begin{bmatrix} -\sin\Theta(u) \\ \cos\Theta(u) \end{bmatrix} + X(0,0) \qquad (3)$$

**Reporting summary**. Further information on research design is available in the Nature Research Reporting Summary linked to this article.

## Data availability
All source data for the figures is available as Supplementary Data 1.

## Code availability
The *Ce*SnAP source code together with trained CurlNet, example demonstrations, and instruction manual are available on https://github.com/murphylab-Princeton/CeSnAP and ftp://gen-ftp.princeton.edu/CeSnAP/.

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

## Acknowledgements
We thank the *C. elegans* Genetics Center for strains (P40 OD010440) and the Murphy lab for discussion. This work was supported by the Glenn Foundation for Medical Research award to C.T.M. (GMFR CNV1001899) and NIH awards to C.T.M. (grant #1RF1AG057341 (NIA), 5R01AG034446 (NIA), and 5DP1GM119167 (NIGMS)). C. T.M. is the Director of the Glenn Center for Aging Research at Princeton and an HHMI-Simons Faculty Scholar. D.E.M. was supported by the Ruth L. Kirschstein NRSA (NIA F32AG062036).

## Author contributions
S.S., D.E.M., and C.T.M. designed experiments. S.S., D.E.M., R.K., and W.K. performed experiments and analyzed data. S.S., D.E.M., and C.T.M wrote the manuscript.

## Competing interests
The authors declare the following competing interests: the method used for automated quantification of curling motor behavior was filed under Provisional Patent # 62/989,317 titled 'Novel High-Throughput Screening Method for Parkinson's Phenotypes Using *C. elegans*.
