## [Peer Review File · Communications Biology]

Reviewers' comments:

Reviewer #1 (Remarks to the Author):

The paper "High-throughput behavioural screen in *C. elegans* reveals novel Parkinson's disease drug candidates" describes a high-throughput assay to study the curling behaviour of *C. elegans*. The authors used this new assay to screen 50 FDA-approved drugs and identified four candidates for potential late-in-life interventions for Parkinson disease.

The paper is well written and includes all relevant information to understand and re-build the proposed screening strategy. The general idea to use curling behaviour as an assay for PD is highly innovative and the preliminary results are very promising. The linking between BCAT1 with PD movement disorder of *C. elegans* (published earlier by the authors) has indeed great potential for further investigations. Since the focus of this work is however the methodological screening assay I will focus on the technical details of this assay in the following.

Major comments:

The technical development of the new screening assays is mainly motivated by two observations (cf. p.3, end of first paragraph): (1) existing approaches rely on (potentially high frame-rate) videos resulting in a heavy data overhead and (2) automatic posture algorithms are not tuned towards curl detection.

Observation (2) is not backed by the literature. In fact there is dedicated work on quantifying curls (or coils; see for example: Broekmans, Onno D., et al. "Resolving coiled shapes reveals new reorientation behaviors in *C. elegans*." *Elife* 5 (2016): e17227.).

Disadvantage (1) is mitigated by using image stills instead of continuous videos. From a technical point of view, all algorithms, which utilise single frames to identify the posture can then be used, so that the frame rate is just reflecting the temporal resolution of the behavioural quantification. Moreover, *C. elegans* can undergo relatively quick posture changes so that high-frame rates might be necessary to avoid under-sampling.

Another disadvantage of using sparsely sampled stills instead of videos is that subtle phenotypes, which might show only small behavioural abbreviations, cannot be captured. The ability to be sensitive to less pronounced behavioural manifestations is however essential in order to be a more versatile screening approach. Therefore, the frame-rate used for a particular screen is always a trade-off and application / assay specific.

In order to serve as a versatile tool the authors therefore need to emphasise the advantages and - more importantly - the limitations of their system with respect to other behavioural research done for *C. elegans*.

My second major technical criticism refers to the custom quantification software itself. The proposed methodology is neither new nor advanced and heavily relies on multiple thresholds. For example, the two filtering strategies to disambiguate worms from artefacts (and colliding / unreasonable entities) is in essence the same as used by most other behavioural quantification tools (image filtering + size and grey value thresholding).

More importantly, the algorithm used to quantify curls also relies on thresholds in order to categorise this type of behaviour in three distinct groups (non-curved; near-curved, curled). Apart from the fact that thresholds result in biased motives (depending on the chosen values) and are genuinely error prone, the discretisation itself can also be problematic. In fact, the use of machine vision algorithms for behavioural quantifications enables to go beyond subjective man-made categories. For example Szigeti et al. have demonstrated that threshold-based categories in fact hide the continuity between behavioural states (see: Szigeti, Balázs, Ajinkya Deogade, and Barbara Webb. "Searching for motifs in the behaviour of larval *Drosophila melanogaster* and *Caenorhabditis elegans* reveals continuity between behavioural states." *Journal of The Royal Society Interface* 12.113 (2015): 20150899.).

The subjectivity of threshold-induced motif quantifications is for example apparent in Figure 1e: For example, the distinction between animal 10 (second row) and animal 14 (fourth row) into curled and near-curved respectively appears arbitrary. The same is true for animal 3 (row 6) and animal 10 (row 3): #3 appears to have an even more pronounced coiled angle than #10, since the body of #3 however partially overlaps the head is not touching the tail so that the shape does not meet the circularity thresholds. Animal 2 (row 6) could also be identified as "near-curved" by domain experts (it is identified to be non-curved by the proposed algorithm), again highlighting the subjectivity of the proposed method.

Instead of using threshold-based categories I would therefore suggest to use continuous curvature values along the spine of the animal to characterise the posture. These values can then be used directly in the statistics.

Finally, I'm wondering if there were attempts to identify animals in front of a less clean background by means of more advanced algorithms. As can be seen in Figure 1c the preparation (10min) and computational analysis (25min) are still the bottle-neck of the proposed procedure. Both could be accelerated by using more tolerant and more reliable algorithmic solutions. For example, recent work has demonstrated that machine learning algorithms can be used to reliably resolve colliding animals, even from image stills (e.g. Klemm, et al. "Deep Distance Transform to Segment Visually Indistinguishable Merged Objects." *GCPR*, 2018).

Minor comments:

the relative numbers on p.6 do not sum to 81% ($63+11+6 = 80$)

why are there 1,050,000 detected objects after filtering? I guess: 32,000 worms * 5 * 7 stills? Please be a bit more specific.

In summary, the paper introduces a very interesting idea of using high-throughput behavioural screens to screen for PD. I'm however not convinced that the used methodology (image stills; discrete behavioural motives; image processing) is meeting the state-of-the-art of other approaches so that I recommend this paper for major revision.

Reviewer #2 (Remarks to the Author):

Review of Sohrabi et al

"High-throughput behavioral screen in *C. elegans* reveals novel Parkinson's disease drug candidates"

Summary:

Parkinson's disease is a neurodegenerative disorder that is associated with the loss of dopaminergic neurons and accumulation of a-synuclein containing aggregates. Recent work from these authors showed that branched-chain amino acid transferase 1 (BCAT1) expression is decreased in Parkinson's patients and that *bcat-1*(RNAi) in *C. elegans* is sufficient to cause spasm-like curling behavior. This work establishes *bcat-1*(RNAi) as a model for investigating the motor control symptoms associated with Parkinson's disease. In this manuscript, Sohrabi et al develop an automated high-throughput assay to detect "curling" in *C. elegans*, which is a behavior that is associated with Parkinson's disease. Using this assay, Sohrabi and team screen FDA approved therapeutics to find those that suppress the spasm-like curling behavior observed in *bcat-1*(RNAi). Sohrabi et al identify four candidates, including one strong candidate, metformin, and implicate metabolic pathways as potential early contributing factors to the onset of neurodegeneration. This manuscript demonstrates that CeSnAP is a useful tool to speed discovery efforts by allowing for rapid behavioral screening in large-scale.

This is a short, methods-based paper consisting of only two figures and one supplemental figure. In the first figure, the authors compare the standard manual assay (12.5 hrs) to their newly developed high-throughput CeSnAP enabled method (25 mins), which marks a 20-fold improvement. The method is straightforward to perform and the custom software allows the user to adjust the filtering parameters to best capture individual worm objects. The software provides color overlays to alert the user to preliminary detection and allows users to over-ride the computer's classification, in the event of an error. In addition, statistical analysis can be performed across all conditions. Direct comparison of the manual and automated methods shows that automated analysis of snapshots is sufficient to capture the same trend as 30s manually annotated movies. Using the automated assay, the authors perform a proof of concept screen of 50 FDA drugs, spanning a range of mechanisms of action. In figure 2, they detail the results of this screen, and perform cursory follow-up analysis of their top hit.

Overall, this method is valuable and has incredible promise for quick screening of behavioral assays. Indeed, this work has yielded some interesting and important results with respect to Parkinson's disease. The paper is generally written clearly, but there are some components that are not discussed in sufficient detail- particularly, rationale for the approaches ultimately taken, and potential limitations and drawbacks of those choices, pitfalls to avoid in this approach, etc. This paper could be fleshed out to make it more of a resource for other labs considering implementing their technique and software. At this time, I would recommend sending it back to the reviewers for revision.

Specific Comments:

1. In Figure 1F, manual scoring is tracking % time spent curling, with ~100 data points, whereas the automated scoring is tracking the % of worms in curled or near-curved position, with ~10 points. Why not score them using the same metric? It seems the automated metric could easily be applied to the manual data to allow direct comparison. If not, please note in text that these are two different measures. Also, is it fair to lump in 'near-curved' when these would not be counted in a manual assay (and ultimately are not used for the automated assay)?

2. I would expect some loss in sensitivity, but I am surprised by the 2-3-fold reduction in sensitivity (6.3 to 2.5 ratio) by switching to snapshots (figure 1G). While 2.5 fold is still sufficient for screening, this is a fairly sizable reduction in the ability of this assay to distinguish between control and queried condition-- noise may be harder to contend with in larger datasets. I think a bit more discussion in the section is warranted for the reader who might be considering setting up such an assay. What are the drawbacks to trying to perform 30s movies in each well instead of snapshots (30s x 96wells = ~50

minutes seems feasible, for example).

3. 63% of worms were identified as successfully found, with 19% immediately triaged due to entanglement and overlap and the other 17% censored out by computer or human intervention. This begs for more discussion in the methods around seeding density within the plate. Presumably over-seeding would result in high overlap and entanglement and this factor would need to be carefully controlled across all conditions. Do you have data on this? It would be straightforward to do a dilution curve to document this and would be a useful thing to include in the supplement for others who might attempt this protocol.

4. The other related consideration is that behavioral changes associated with RNAi condition or drug treatment could result in more or less clumping and entanglement. Rather than pooling all data and discussing the discarded percentages as a whole, I would like to know whether there is a significant difference in the excluded data in the control vs. *bcat-1* conditions. In other words, is it possible this method over or under represents actual behavioral phenotypes?

5. In Panel 2C the results of the 50 compound screen are summarized. The variation between replicates is striking. The authors should discuss the source of this variation and discuss their criteria for identifying hit candidate compounds amid this noise.

6. Is Ethosuximide considered a positive control for this study? If so, I would recommend an overt statement to that effect and some indication on the figure itself. If not, can you include PD medications prescribed for motor symptoms (selegiline (seleg.), trihexyphenidyl (trihex.)) as positive controls here for reference so this article stands alone from the companion article.

7. There is no discussion or analysis of the drugs that make the curling phenotype worse. Certainly Doxylamine (antihistamine) appears to cause significant worsening. Further, Vecuronium and chlorzoxazone, both muscle relaxants (which seem likely to be prescribed to treat spasms) also appear to worsen (on average) the outcome. Was there any validation of these findings?

8. Because lab websites change and are not always well maintained, the source code, tutorial and demo material should be placed in a public repository (GitHub, etc). The current link in the text take the reader to the Murphy Lab. The actual materials are a few clicks away (<ftp://gen-ftp.princeton.edu/CeSnAP/>).

9. Instructions for setting up CeSnAP should accompany this article as supplement.

10. I find figure 2B to be confusing. It is clear that the authors are including experimental information that is in the accompanying manuscript, however, it makes it difficult to understand what they are trying to summarize in this manuscript. Perhaps they can gray-out the portions of this that do not apply to the work described within this manuscript. Do the numbers in the center include other experimental approaches? If so, why are these being pooled. What is meant by "experiment" and "condition" in this regime?

11. Minor comment- If the colors in Figure 2D are not meaningful (with respect to the colors in 2C) I would recommend switching to grayscale. My instinct is to link the colors between the 2B and 2D, but they appear to be unrelated to each other.

12. How do you deal with the presence of progeny? There must be young worms on the plate that get washed and aliquoted with the older adults- are you taking some extra measures to eliminate

the younger worms? How are you staging your worms for the automated assay?

Reviewer #3 (Remarks to the Author):

In the paper entitled "High-throughput behavioral screen in *C. elegans* reveals novel Parkinson's disease drug candidates", Sohrabi and colleagues present a new method to rapidly analyze the curling phenotype in *C. elegans*. Using this approach they were able to screen for 400 conditions, including 50 drugs, 20 times faster than before and to identify 4 very promising hits for PD.

The paper is very well written, focused and it presents a very useful approach for the *C. elegans* community. This smart approach allows to quantify a subtle locomotion defect, to save time and to perform unbiased screenings. The method is applicable to PD related projects but also, more in general, to screening related to the curling phenotype, which involves multiple neurons. Moreover, they identify leading hits for mitochondria related neurodegeneration, among which metformin is the most interesting.

I have some major (meaning there is a need for more experiments) and minor (meaning there is a need for text changes) comments.

Major comments:

1) It is puzzling (and the very first time) to review a paper and a "companion" one, which is in a preprint version. While I understand the need to publish the technique separately to the application and all the results, I don't understand how data shared among two papers have to be commented/reviewed. For example, in Fig. 1h and 2b data presented are neither explained nor showed, referring to Mor et al. 2020. However, in Mor et al., 2020 I could not find any mentioning to CeSNAP or to so many conditions/experiments. If Sohrabi et al. manuscript has to be a self-standing paper, I think these sunburst plots have to be changed and refer only to the data presented here, e.g. the drug screening. Or, alternatively, please explain which are the 400 conditions tested in 75 experiments and all other information showed in the plots.

2) Since this is mainly a technical paper, I think it would be important to deepen the protocol for other users and try to improve the consistency: so why 7 snapshots were performed? Why not more or less? The authors should test different numbers of snapshots (e.g. 5-10-20) and the change in consistency/variability/reliability versus time spent/memory load, should be discussed. In fact, from fig. 2c is clear that a large variability is obtained, which might be improved increasing the number of snapshots.

3) One out of five best hits (felodipine), showed to be a false positive, which is something expected. However false negatives are extremely dangerous. Indeed, among the negative ones some have been showed to be neuroprotective in PD models (e.g. prednisone and valproic acid). Moreover, valproate is used with ethosuximide for epilepsy and convulsion treatment. To test for a "false negatives rate" I suggest a post-screen validation of a sampling of at least 5 negative drugs, among those known to be protecting from neurodegeneration in PD models.

4) A control that the 4 best hits are not influencing the RNAi machinery and therefore its effects in *bcat-1* silencing is needed. For example, silence *unc-54* or *mom-2* and check for the expected phenotypes after treating with drugs.

Minor comments:

- 5) There is already one paper counting the number of curls in *C. elegans* mutants, in automated fashion (Hoshi and Shingai, 2006). The authors should add a reference to it and briefly comment advantages and disadvantages of their approach compared to this other one.
- 6) It is important to discuss more the two phenotypes: curled and near-curved positions. Are they part of the same defect or two different manifestations that can be separated? How much are they contributing to the overall sum in all conditions (as shown in 1h)? How much the rescue with drugs is impacting on each one?
- 7) In panels 1h and 2b testing of 400 different unknown conditions is described. Taking in account comment 1, how long did it take to screen 400 conditions is an important information to be added.
- 8) Some drugs show an interesting enhancing phenotype, up to 140% of control. This is an important information for twofold screenings. The author should discuss this aspect.
- 9) In Mat. and Meth. a long list of strains is listed, which I don't get when were used. Please correct or explain their use.
- 10) It is not clear to me how the snapshots were taken and the technical setup used for the automated acquisition with the motorized microscope. Every how many minutes the same well is photographed again? Please explain in order to allow any researcher to reproduce the work.
- 11) There is an evident discrepancy between panel 1f and 1g, which I understand are the same experiment performed on different days. Please discuss it.
- 12) The curly phenotype is an age related one, hence the 8 days after adulthood, which make sense being related to PD. Please make it explicit.
- 13) The comment to panel 2A (Furthermore, we found that using snapshots to quantify the curled position distinguished between *bcat-1*(RNAi) and control groups more effectively than the near-curved position) is neither clear nor validated with statistics. Please explain and complete. Moreover, I would move it to Figure 1.
- 14) If panel 1h is going to be maintained it should be moved to Fig. 2.
- 15) From Yao et al., 2018 I understand that the reduction of *bcat-1* in *C. elegans* enhances α -synuclein effects and does not by itself promote dopaminergic neurodegeneration, as stated in page 2. Please correct.

Review of Sohrabi et al

“High-throughput behavioral screen in *C. elegans* reveals novel Parkinson’s disease drug candidates”

Summary:

Parkinson’s disease is a neurodegenerative disorder that is associated with the loss of dopaminergic neurons and accumulation of α -synuclein containing aggregates. Recent work from these authors showed that branched-chain amino acid transferase 1 (*BCAT1*) expression is decreased in Parkinson’s patients and that *bcat-1(RNAi)* in *C. elegans* is sufficient to cause spasm-like curling behavior. This work establishes *bcat-1(RNAi)* as a model for investigating the motor control symptoms associated with Parkinson’s disease. In this manuscript, Sohrabi et al develop an automated high-throughput assay to detect “curling” in *C. elegans*, which is a behavior that is associated with Parkinson’s disease. Using this assay, Sohrabi and team screen FDA approved therapeutics to find those that suppress the spasm-like curling behavior observed in *bcat-1(RNAi)*. Sohrabi et al identify four candidates, including one strong candidate, metformin, and implicate metabolic pathways as potential early contributing factors to the onset of neurodegeneration. This manuscript demonstrates that CeSnAP is a useful tool to speed discovery efforts by allowing for rapid behavioral screening in large-scale.

This is a short, methods-based paper consisting of only two figures and one supplemental figure. In the first figure, the authors compare the standard manual assay (12.5 hrs) to their newly developed high-throughput CeSnAP enabled method (25 mins), which marks a 20-fold improvement. The method is straightforward to perform and the custom software allows the user to adjust the filtering parameters to best capture individual worm objects. The software provides color overlays to alert the user to preliminary detection and allows users to over-ride the computer’s classification, in the event of an error. In addition, statistical analysis can be performed across all conditions. Direct comparison of the manual and automated methods shows that automated analysis of snapshots is sufficient to capture the same trend as 30s manually annotated movies. Using the automated assay, the authors perform a proof of concept screen of 50 FDA drugs, spanning a range of mechanisms of action. In figure 2, they detail the results of this screen, and perform cursory follow-up analysis of their top hit.

Overall, this method is valuable and has incredible promise for quick screening of behavioral assays. Indeed, this work has yielded some interesting and important results with respect to Parkinson’s disease. The paper is generally written clearly, but there are some components that are not discussed in sufficient detail- particularly, rationale for the approaches ultimately taken, and potential limitations and drawbacks of those choices, pitfalls to avoid in this approach, etc. This paper could be fleshed out to make it more of a resource for other labs considering implementing their technique and software. At this time, I would recommend sending it back to the reviewers for revision.

Specific Comments:

1. In Figure 1F, manual scoring is tracking % time spent curling, with ~100 data points, whereas the automated scoring is tracking the % of worms in curled or near-curved position, with ~10 points. Why not score them using the same metric? It seems the automated metric could easily be

applied to the manual data to allow direct comparison. If not, please note in text that these are two different measures. Also, is it fair to lump in 'near-curling' when these would not be counted in a manual assay (and ultimately are not used for the automated assay)?

2. I would expect some loss in sensitivity, but I am surprised by the 2-3-fold reduction in sensitivity (6.3 to 2.5 ratio) by switching to snapshots (figure 1G). While 2.5 fold is still sufficient for screening, this is a fairly sizable reduction in the ability of this assay to distinguish between control and queried condition-- noise may be harder to contend with in larger datasets. I think a bit more discussion in the section is warranted for the reader who might be considering setting up such an assay. What are the drawbacks to trying to perform 30s movies in each well instead of snapshots (30s x 96wells = ~50 minutes seems feasible, for example).
3. 63% of worms were identified as successfully found, with 19% immediately triaged due to entanglement and overlap and the other 17% censored out by computer or human intervention. This begs for more discussion in the methods around seeding density within the plate. Presumably over-seeding would result in high overlap and entanglement and this factor would need to be carefully controlled across all conditions. Do you have data on this? It would be straightforward to do a dilution curve to document this and would be a useful thing to include in the supplement for others who might attempt this protocol.
4. The other related consideration is that behavioral changes associated with RNAi condition or drug treatment could result in more or less clumping and entanglement. Rather than pooling all data and discussing the discarded percentages as a whole, I would like to know whether there is a significant difference in the excluded data in the control vs. *bcat-1* conditions. In other words, is it possible this method over or under represents actual behavioral phenotypes?
5. In Panel 2C the results of the 50 compound screen are summarized. The variation between replicates is striking. The authors should discuss the source of this variation and discuss their criteria for identifying hit candidate compounds amid this noise.
6. Is Ethosuximide considered a positive control for this study? If so, I would recommend an overt statement to that effect and some indication on the figure itself. If not, can you include PD medications prescribed for motor symptoms (selegiline (seleg.), trihexyphenidyl (trihex.)) as positive controls here for reference so this article stands alone from the companion article.
7. There is no discussion or analysis of the drugs that make the curling phenotype worse. Certainly Doxylamine (antihistamine) appears to cause significant worsening. Further, Vecuronium and chlorzoxazone, both muscle relaxants (which seem likely to be prescribed to treat spasms) also appear to worsen (on average) the outcome. Was there any validation of these findings?
8. Because lab websites change and are not always well maintained, the source code, tutorial and demo material should be placed in a public repository (GitHub, etc). The current link in the text take the reader to the Murphy Lab. The actual materials are a few clicks away (<ftp://gen-ftp.princeton.edu/CeSnAP/>).

9. Instructions for setting up CeSnAP should accompany this article as supplement.
10. I find figure 2B to be confusing. It is clear that the authors are including experimental information that is in the accompanying manuscript, however, it makes it difficult to understand what they are trying to summarize in *this* manuscript. Perhaps they can gray-out the portions of this that do not apply to the work described within this manuscript. Do the numbers in the center include other experimental approaches? If so, why are these being pooled. What is meant by “experiment” and “condition” in this regime?
11. Minor comment- If the colors in Figure 2D are not meaningful (with respect to the colors in 2C) I would recommend switching to grayscale. My instinct is to link the colors between the 2B and 2D, but they appear to be unrelated to each other.
12. How do you deal with the presence of progeny? There must be young worms on the plate that get washed and aliquoted with the older adults- are you taking some extra measures to eliminate the younger worms? How are you staging your worms for the automated assay?

We would like to thank the editor and reviewers for their valuable comments and suggestions. As a result of their feedback, we have completely re-designed the CeSnAP algorithm and incorporated a machine learning approach. Please see below for a point-by-point description of all changes.

Reviewer #1 (Remarks to the Author):

The paper "High-throughput behavioural screen in C. elegans reveals novel Parkinson's disease drug candidates" describes a high-throughput assay to study the curling behaviour of C. elegans. The authors used this new assay to screen 50 FDA-approved drugs and identified four candidates for potential late-in-life interventions for Parkinson disease.

The paper is well written and includes all relevant information to understand and re-build the proposed screening strategy. The general idea to use curling behaviour as an assay for PD is highly innovative and the preliminary results are very promising. The linking between BCAT1 with PD movement disorder of C. elegans (published earlier by the authors) has indeed great potential for further investigations. Since the focus of this work is however the methodological screening assay I will focus on the technical details of this assay in the following.

We thank the reviewer for their positive feedback.

Major comments:

The technical development of the new screening assays is mainly motivated by two observations (cf. p.3, end of first paragraph): (1) existing approaches rely on (potentially high frame-rate) videos resulting in a heavy data overhead and (2) automatic posture algorithms are not tuned towards curl detection.

Observation (2) is not backed by the literature. In fact there is dedicated work on quantifying curls (or coils; see for example: Broekmans, Onno D., et al. "Resolving coiled shapes reveals new reorientation behaviors in C. elegans." Elife 5 (2016): e17227.).

The reviewer's second observation is valid since we did not discuss these matters in the original manuscript. There are powerful trackers than can quantify *C. elegans* body shape and locomotion through coiling, entanglement, and omega bends (Broekmans et al., 2016; Roussel et al., 2014). Although robust and advanced, their final results require user inspection and approval. Inspecting

worms' track history or checking each video may take one to a few hours in a curling assay with 7 conditions (>700 worms). Our newly re-designed algorithm can now accurately quantify multiple distinct Coiled, Curled, Near-curved, and Non-curved postures, while eliminating the need for any user interaction. In addition to the low data overhead of using snapshots instead of videos, our user-free automated quantification workflow represents a marked advance over existing curling/coiling detection systems.

Disadvantage (1) is mitigated by using image stills instead of continuous videos. From a technical point of view, all algorithms, which utilize single frames to identify the posture can then be used, so that the frame rate is just reflecting the temporal resolution of the behavioural quantification. Moreover, C. elegans can undergo relatively quick posture changes so that high-frame rates might be necessary to avoid under-sampling.

Another disadvantage of using sparsely sampled stills instead of videos is that subtle phenotypes, which might show only small behavioural abbreviations, cannot be captured. The ability to be sensitive to less pronounced behavioural manifestations is however essential in order to be a more versatile screening approach. Therefore, the frame-rate used for a particular screen is always a trade-off and application / assay specific.

In order to serve as a versatile tool the authors therefore need to emphasise the advantages and - more importantly - the limitations of their system with respect to other behavioural research done for C. elegans.

We agree that choosing a frame rate always presents a trade-off, and must be tailored to the specific assay and phenotype of interest. CeSnAP is versatile in that it can analyze both videos and snapshots, and these applications can serve multiple, diverse purposes. Analysis of videos with CeSnAP involves extracting stills at any preferred time interval, and can therefore achieve sufficiently high frame rates to detect subtle phenotypes. Therefore, users of CeSnAP can apply the program to the identification of quick and/or subtle movement changes, if desired.

We have also introduced an easy-to-use workflow for training and classification of worm images. The user simply needs to extract mask images using SnapSegment, then train a neural network based on their desired classification using SnapTrain. Trained networks can be loaded onto SnapMachine and used later to score new experiments with high reliability. The application of this workflow is not limited

to curling analysis and can be utilized for scoring images from other types of assays that rely on worm posture, resulting in greater versatility of the software.

The use of CeSnAP with our automated snapshot capturing setup can serve the purpose of high-throughput detection and screening of robust movement changes. This system uses stills captured for each well at 30s-2min intervals, which we have demonstrated is sufficient to rapidly screen for chemicals that ameliorate the severe curling phenotype. Our drug screen provides proof-of-concept that this snapshot approach maintains sufficient resolution while reducing data overhead, for the purpose of identifying new potential therapeutics for Parkinson's disease. We have also carefully revised the manuscript to include additional discussion on advantages and limitations of our new methodology with respect to other behavioral research done with *C. elegans*.

My second major technical criticism refers to the custom quantification software itself. The proposed methodology is neither new nor advanced and heavily relies on multiple thresholds. For example, the two filtering strategies to disambiguate worms from artefacts (and colliding / unreasonable entities) is in essence the same as used by most other behavioural quantification tools (image filtering + size and grey value thresholding).

*More importantly, the algorithm used to quantify curls also relies on thresholds in order to categorise this type of behaviour in three distinct groups (non-curved; near-curved, curled). Apart from the fact that thresholds result in biased motives (depending on the chosen values) and are genuinely error prone, the discretisation itself can also be problematic. In fact, the use of machine vision algorithms for behavioural quantifications enables to go beyond subjective man-made categories. For example Szigeti et al. have demonstrated that threshold-based categories in fact hide the continuity between behavioural states (see: Szigeti, Balázs, Ajinkya Deogade, and Barbara Webb. "Searching for motifs in the behaviour of larval *Drosophila melanogaster* and *Caenorhabditis elegans* reveals continuity between behavioural states." *Journal of The Royal Society Interface* 12.113 (2015): 20150899.).*

The subjectivity of threshold-induced motif quantifications is for example apparent in Figure 1e: For example, the distinction between animal 10 (second row) and animal 14 (forth row) into curled and near-curved respectively appears arbitrary. The same is true for animal 3 (row 6) and animal 10 (row 3): #3 appears to have an even more pronounced coiled angle than #10, since the body of #3 however partially overlaps the head is not touching the tail so that the shape does not meet the circularity thresholds. Animal 2 (row 6) could also be identified as "near-curved" by domain experts (it is identified to be non-

curled by the proposed algorithm), again highlighting the subjectivity of the proposed method. Instead of using threshold-based categories I would therefore suggest to use continuous curvature values along the spine of the animal to characterise the posture. These values can then be used directly in the statistics.

We thank the reviewer for these comments. In light of this feedback, we have completely re-designed the CeSnAP algorithm as suggested by the reviewer. Although the original version of CeSnAP successfully quantified curling and automated a previously labor-intensive assay, we agree that the methodology based on thresholding techniques introduces subjectivity and potential error. **Therefore, we have developed a machine learning (ML) algorithm to go beyond subjective man-made categories.** CeSnAP now extracts worm images from snapshots and uses thousands of carefully categorized mask images to train a convolutional neural network, from which worm postures of new experiments can then be automatically detected and classified without user-supervision. Additionally, CeSnAP now uses continuous curvature values along the spine of the animal to model the worm body; for example, CeSnAP is able to output length and area of more than 700 worms with only 20mins of benchwork.

Finally, I'm wondering if there were attempts to identify animals in front of a less clean background by means of more advanced algorithms. As can be seen in Figure 1c the preparation (10min) and computational analysis (25min) are still the bottle-neck of the proposed procedure. Both could be accelerated by using more tolerant and more reliable algorithmic solutions. For example, recent work has demonstrated that machine learning algorithms can be used to reliably resolve colliding animals, even from image stills (e.g. Klemm, et al. "Deep Distance Transform to Segment Visually Indistinguishable Merged Objects." GCPR, 2018).

As correctly pointed out by the reviewer, our previous methodology for computational analysis was the bottle-neck of our proposed procedure. Utilizing our new ML algorithm addresses this issue. We have now developed a more tolerant and more reliable method to detect and score animals which only requires 2-3 minutes of the user's time to set up and export the results. The accuracy of worm classification for curling analysis is more than 95%, thereby eliminating the need for user post-inspection. In addition, a series of screenshots from SnapMachine during detection and classification is included in the supplementary material to showcase its performance in front of a less clean background (Supplementary Figure 2-3). Overall, the reviewer's comments have helped us to develop a truly autonomous algorithm.

Minor comments:

the relative numbers on p.6 do not sum to 81% (63+11+6 = 80)

We apologize for this mistake, which was due to a rounding error. The revised manuscript has been carefully reviewed to avoid this type of error.

why are there 1,050,000 detected objects after filtering? I guess: 32,000 worms × 5 × 7 stills? Please be a bit more specific.

The calculation for number of detected objects is almost correct. The rounds of snapshots during our experiments have ranged from 4 to 9. Also, 1,050,000 represented the number of detected objects after first-pass filtration in our old threshold-based CeSnAP. As the manuscript is modified to only include the drug screening and method optimization data unique to this paper, these numbers have been updated accordingly. We have made every effort to be more specific and avoid ambiguous descriptions in the revised manuscript.

In summary, the paper introduces a very interesting idea of using high-throughput behavioural screens to screen for PD. I'm however not convinced that the used methodology (image stills; discrete behavioural motives; image processing) is meeting the state-of-the-art of other approaches so that I recommend this paper for major revision.

We have carried out the reviewer's recommended major changes to the algorithm, which now uses a ML approach and continuous curvature values to characterize worm posture. We believe these changes have greatly improved the power and versatility of the algorithm, and we hope that our new methodology now meets the state-of-the-art of other approaches.

Reviewer #2 (Remarks to the Author):

Review of Sohrabi et al

"High-throughput behavioral screen in C. elegans reveals novel Parkinson's disease drug candidates"

Summary:

*Parkinson's disease is a neurodegenerative disorder that is associated with the loss of dopaminergic neurons and accumulation of α -synuclein containing aggregates. Recent work from these authors showed that branched-chain amino acid transferase 1 (BCAT1) expression is decreased in Parkinson's patients and that *bcat-1(RNAi)* in *C. elegans* is sufficient to cause spasm-like curling behavior. This work*

establishes bcat-1(RNAi) as a model for investigating the motor control symptoms associated with Parkinson's disease. In this manuscript, Sohrabi et al develop an automated high-throughput assay to detect "curling" in C. elegans, which is a behavior that is associated with Parkinson's disease. Using this assay, Sohrabi and team screen FDA approved therapeutics to find those that suppress the spasm-like curling behavior observed in bcat-1(RNAi). Sohrabi et al identify four candidates, including one strong candidate, metformin, and implicate metabolic pathways as potential early contributing factors to the onset of neurodegeneration. This manuscript demonstrates that CeSnAP is a useful tool to speed discovery efforts by allowing for rapid behavioral screening in large-scale.

This is a short, methods-based paper consisting of only two figures and one supplemental figure. In the first figure, the authors compare the standard manual assay (12.5 hrs) to their newly developed high-throughput CeSnAP enabled method (25 mins), which marks a 20-fold improvement. The method is straightforward to perform and the custom software allows the user to adjust the filtering parameters to best capture individual worm objects. The software provides color overlays to alert the user to preliminary detection and allows users to over-ride the computer's classification, in the event of an error. In addition, statistical analysis can be performed across all conditions. Direct comparison of the manual and automated methods shows that automated analysis of snapshots is sufficient to capture the same trend as 30s manually annotated movies. Using the automated assay, the authors perform a proof of concept screen of 50 FDA drugs, spanning a range of mechanisms of action. In figure 2, they detail the results of this screen, and perform cursory follow-up analysis of their top hit.

Overall, this method is valuable and has incredible promise for quick screening of behavioral assays. Indeed, this work has yielded some interesting and important results with respect to Parkinson's disease. The paper is generally written clearly, but there are some components that are not discussed in sufficient detail- particularly, rationale for the approaches ultimately taken, and potential limitations and drawbacks of those choices, pitfalls to avoid in this approach, etc. This paper could be fleshed out to make it more of a resource for other labs considering implementing their technique and software. At this time, I would recommend sending it back to the reviewers for revision.

We thank the reviewer for their positive feedback. We have now significantly revised the algorithm to include machine learning, and have provided in-depth discussion and analysis of both the advantages and limitations of our new methodology in the context of existing technologies. We also revised the manuscript significantly to make it a resource for other labs. We hope that these changes have improved the utility and accessibility of our method.

Specific Comments:

1. In Figure 1F, manual scoring is tracking % time spent curling, with ~100 data points, whereas the automated scoring is tracking the % of worms in curled or near-curved position, with ~10 points. Why not score them using the same metric? It seems the automated metric could easily be applied to the manual data to allow direct comparison. If not, please note in text that these are two different measures. Also, is it fair to lump in 'near-curved' when these would not be counted in a manual assay (and ultimately are not used for the automated assay)?

The manual and automated analyses are two different measures, and we have now clarified this in the manuscript. The manual assay measures % of time spent curling for each individual worm, resulting in ~100 data points (1 data point per worm). In the automated analysis, the program instead measures the % of worms that are curling per individual well, with 10-15 worms per well, resulting in ~10 data points (1 data point per well). However, to allow for more direct comparison, we have now collapsed the manual data such that each data point represents a curling value per well rather than per worm (Figure 2b), which has not changed any of our conclusions.

The general procedure for manual hand-counting is to start the timer as an animal starts to curl. This necessarily includes time when the animal is near-curved, prior to being fully coiled (Supplementary Video 1). Therefore, to best compare manual and automated measurements, we originally used Curled + Near-curved values for the automated analysis. Our current program can accurately discriminate a greater diversity of postures, including fully coiled, curled, and near-curved states. Thus, in our revised comparison of manual versus automated measurements, the automated analysis now uses Coiled + Curled + Near-curved values, in order to best approximate the manual scoring criteria. We have also extensively discussed the significance of these postures in the revised manuscript.

2. I would expect some loss in sensitivity, but I am surprised by the 2-3-fold reduction in sensitivity (6.3 to 2.5 ratio) by switching to snapshots (figure 1G). While 2.5 fold is still sufficient for screening, this is a fairly sizable reduction in the ability of this assay to distinguish between control and queried condition-- noise may be harder to contend with in larger datasets. I think a bit more discussion in the section is warranted for the reader who might be considering setting up such an assay.

We have followed the reviewer's suggestion and extensively discussed the reason for this sizable reduction in sensitivity in the revised manuscript.

What are the drawbacks to trying to perform 30s movies in each well instead of snapshots (30s x 96wells = ~50 minutes seems feasible, for example).

Worms in liquid alternate between swimming and quiescence (Ghosh & Emmons, 2008; Ikeda et al., 2020). The quiescence state grows longer with time after transfer into liquid buffer, which diminishes the difference between control and *bcat-1* curling levels (Figure 2e). Thus, for 30s videos, animals must be picked immediately prior to filming each video and cannot be washed into a 96-well plate all at once. This severely limits the throughput of video-based approaches. Our snapshot approach offers an alternative to videos that allows for high-throughput screening as we have demonstrated.

3. 63% of worms were identified as successfully found, with 19% immediately triaged due to entanglement and overlap and the other 17% censored out by computer or human intervention. This begs for more discussion in the methods around seeding density within the plate. Presumably over-seeding would result in high overlap and entanglement and this factor would need to be carefully controlled across all conditions. Do you have data on this? It would be straightforward to do a dilution curve to document this and would be a useful thing to include in the supplement for others who might attempt this protocol.

In the revised manuscript, manual censoring was completely eliminated since the machine learning neural network provides a fully autonomous detection and classification workflow. This high reliability comes at the cost of slightly higher overall censoring rate (37% for the older version of CeSnAP compared to 46.11% in ML-based CeSnAP), which can be compensated for by increasing the number of snapshots. Following the reviewer's suggestion, we have added a more detailed discussion of the distribution of worms across wells and a dilution analysis (Supplementary Figure 1a(bars)). Our fast loading technique can distribute worms across wells near-evenly (Supplementary Figure 1a (dashed line)). Furthermore, over-seeding, as correctly stated, would result in high overlap and lower detection rate accordingly (Supplementary Figure 1b). Our high-throughput assay works optimally with 50-150 worms per condition. A relevant discussion of this point has been added to our revised manuscript.

*4. The other related consideration is that behavioral changes associated with RNAi condition or drug treatment could result in more or less clumping and entanglement. Rather than pooling all data and discussing the discarded percentages as a whole, I would like to know whether there is a significant difference in the excluded data in the control vs. *bcat-1* conditions. In other words, is it possible this method over or under represents actual behavioral phenotypes?*

To investigate whether behavioral changes associated with RNAi condition could potentially result in more censoring of worms, we have plotted out % censored worms in 8 different experiments for control and *bcat-1(RNAi)* (Figure 1 for Reviewers). We found that there is no significant difference in the excluded data in the control vs. *bcat-1(RNAi)* conditions, suggesting that the exclusion criteria are not biased and do not over/under-represent behavioral phenotypes.

Figure 1 for Reviewers: Percentage of worms that were censored from control and *bcat-1(RNAi)* conditions across 8 different experiments shows no significant difference.

5. In Panel 2C the results of the 50 compound screen are summarized. The variation between replicates is striking. The authors should discuss the source of this variation and discuss their criteria for identifying hit candidate compounds amid this noise.

High variability between replicates is an inherent part of this behavioral assay. *bcat1(RNAi)* to control also fluctuate from experiment to experiment. However, despite this variability, *bcat1(RNAi)* consistently shows significantly greater curling than controls. We have generally used 7 or more wells (replicates) to help mitigate the variability. Although three rounds of snapshots for each well (~20 snapshots total) was found to be sufficient to identify hit candidates in our screen, we often capture 5-9 rounds to make sure the results remain consistent across all rounds (Supplementary Figure 1c). In addition to repeating experiments, we have done the many follow-up validation of drug candidates using 30s videos (Figure 2f). Using this workflow, we were able to validate 4 out of 5 top hits (Figure 3c). Overall, the variability in the curling phenotype does not hinder the ability to make meaningful discoveries using this workflow.

6. Is Ethosuximide considered a positive control for this study? If so, I would recommend an overt statement to that effect and some indication on the figure itself. If not, can you include PD medications

prescribed for motor symptoms (selegiline (seleg.), trihexyphenidyl (trihex.)) as positive controls here for reference so this article stands alone from the companion article.

We do not consider Ethosuximide to be a positive control. As per the reviewer's suggestion, we have included PD medications prescribed for motor symptoms in the revised manuscript (Figure 3a); thus our article can stand alone from the companion paper.

7. There is no discussion or analysis of the drugs that make the curling phenotype worse. Certainly Doxylamine (antihistamine) appears to cause significant worsening. Further, Vecuronium and chlorzoxazone, both muscle relaxants (which seem likely to be prescribed to treat spasms) also appear to worsen (on average) the outcome. Was there any validation of these findings?

Since finding potential therapies that may improve PD-related motor symptoms was the focus of this study, drugs that potentially worsen the phenotype were not further pursued. However, our results and the CeSnAP method may open up new avenues of investigation for future studies that may wish to identify factors that worsen curling behavior.

8. Because lab websites change and are not always well maintained, the source code, tutorial and demo material should be placed in a public repository (GitHub, etc). The current link in the text take the reader to the Murphy Lab. The actual materials are a few clicks away (<ftp://gen-ftp.princeton.edu/CeSnAP/>).

The source code and instructions have been placed on GitHub. However, we cannot upload demo materials in public repositories because of the size limitation, and therefore demo materials remain accessible through the Murphy lab website. We have also updated the link so it can direct users straight to CeSnAP materials.

9. Instructions for setting up CeSnAP should accompany this article as supplement.

An instruction manual for CeSnAP has been added as a supplement following the reviewer's suggestion.

10. I find figure 2B to be confusing. It is clear that the authors are including experimental information that is in the accompanying manuscript, however, it makes it difficult to understand what they are trying to summarize in this manuscript. Perhaps they can gray-out the portions of this that do not apply to the work described within this manuscript. Do the numbers in the center include other experimental approaches? If so, why are these being pooled. What is meant by "experiment" and "condition" in this regime?

To stand alone from the accompanying article, the sunburst plots have been changed to refer only to the data described within this manuscript. We use the term “condition” to describe one treatment group, i.e. *bcat-1(RNAi)* worms treated with metformin. The term “experiment” is used to describe a set of multiple conditions tested in parallel, i.e. control worms, *bcat-1(RNAi)* worms treated with vehicle, and *bcat-1(RNAi)* worms treated with metformin (3 total conditions in 1 experiment). The manuscript has been revised to include clear definitions of these terms. In the revised sunburst plots, the numbers in the center represents the number of conditions tested within designated categories (Figure 2g). For instance, we screened 50 FDA-approved drugs, with some drugs having been tested repeatedly in multiple experiments, totaling 94 experiments for these drugs. We hope that these changes showcase more clearly the total # of conditions that we tested which was not possible without a high-throughput platform.

11. Minor comment- If the colors in Figure 2D are not meaningful (with respect to the colors in 2C) I would recommend switching to grayscale. My instinct is to link the colors between the 2B and 2D, but they appear to be unrelated to each other.

We have generally used gray, red, and green for control, *bcat-1*, and drug treatment, respectively. In posture analysis, Red, magenta, green, blue, and yellow represent Coiled, Curled, Near-curved, Non-curved, and Censored mask images. To further address the issue mentioned, the sunburst plots are now in a separate figure from the drug screen results and validation. We hope that these changes eliminate any confusion.

12. How do you deal with the presence of progeny? There must be young worms on the plate that get washed and aliquoted with the older adults- are you taking some extra measures to eliminate the younger worms? How are you staging your worms for the automated assay?

Although the trained machine learning network can filter out L1 to L3 stage progeny, we used a temperature-sensitive mutant strain (*fem-1(hc17); fer-15(b26)*) in which worms were rendered spermless by incubation at 25°C from L2- L4, so there are no progeny in these experiments. This is a common technique to prevent progeny production, and does not affect behavior or lifespan (Murphy et al., 2003).

Reviewer #3 (Remarks to the Author):

In the paper entitled "High-throughput behavioral screen in C. elegans reveals novel Parkinson's disease drug candidates", Sohrabi and colleagues present a new method to rapidly analyze the curling phenotype in C. elegans. Using this approach they were able to screen for 400 conditions, including 50 drugs, 20 times faster than before and to identify 4 very promising hits for PD.

The paper is very well written, focused and it presents a very useful approach for the C. elegans community. This smart approach allows to quantify a subtle locomotion defect, to save time and to perform unbiased screenings. The method is applicable to PD related projects but also, more in general, to screening related to the curling phenotype, which involves multiple neurons. Moreover, they identify leading hits for mitochondria related neurodegeneration, among which metformin is the most interesting.

I have some major (meaning there is a need for more experiments) and minor (meaning there is a need for text changes) comments.

We thank the reviewer for their positive feedback.

Major comments:

1) It is puzzling (and the very first time) to review a paper and a "companion" one, which is in a preprint version. While I understand the need to publish the technique separately to the application and all the results, I don't understand how data shared among two papers have to be commented/reviewed. For example, in Fig. 1h and 2b data presented are neither explained nor showed, referring to Mor et al. 2020. However, in Mor et al., 2020 I could not find any mentioning to CeSNAP or to so many conditions/experiments. If Sohrabi et al. manuscript has to be a self-standing paper, I think these sunburst plots have to be changed and refer only to the data presented here, e.g. the drug screening. Or, alternatively, please explain which are the 400 conditions tested in 75 experiments and all other information showed in the plots.

As suggested, we have significantly revised the manuscript with the goal to be a self-standing article. Thus, the sunburst plots have been modified to only include drug screening and method optimization data unique to this paper (Figure 2g). All data referred to by Mor et al. (PNAS), has been removed from

this manuscript. The Materials and Methods, and all references to the companion article have also been changed accordingly.

2) Since this is mainly a technical paper, I think it would be important to deepen the protocol for other users and try to improve the consistency: so why 7 snapshots were performed? Why not more or less? The authors should test different numbers of snapshots (e.g. 5-10-20) and the change in consistency/variability/reliability versus time spent/memory load, should be discussed. In fact, from fig. 2c is clear that a large variability is obtained, which might be improved increasing the number of snapshots.

We have taken careful measures to deepen the description of the protocol in the revised version of this technical paper. Worms in liquid alternate between swimming and quiescence (Ghosh & Emmons, 2008; Ikeda et al., 2020). The quiescent state grows longer with time after transfer into liquid buffer, which diminishes the difference between control and *bcat-1* curling levels (Figure 2e). We found it beneficial to stimulate worms on a shaker every 15 min to restart movement of worms in a state of quiescence (Figure 1d). We suggest 7 snapshots for each well of up to 7 experimental conditions (and up to 7 wells per condition) to ensure each round finishes in under 15min. However, users can take as many snapshots per round as they see fit. Since image acquisition setup and analyzing software are both automated, increasing the numbers of snapshots won't affect time spent for each experiment. Additionally, CeSnAP can load and process a batch of snapshots at a time which enables it to analyze a large total number of snapshots without memory overload.

High variability between replicates is an inherent part of this behavioral assay. We have generally used 7 or more wells (replicates) to help mitigate the variability. Although three rounds of snapshots for each well (~20 snapshots total) was found to be sufficient to identify hit candidates in our screen, we often capture 5-9 rounds to make sure the results remain consistent across all rounds (Supplementary Figure 1c). In addition to repeating experiments, we have done the many follow-up validation of drug candidates using 30s videos (Figure 2f). Overall, the variability in the curling phenotype does not hinder the ability to make meaningful discoveries using this workflow. Upon the reviewer's suggestion, CeSnAP performance versus distribution of worms (Supplementary Figure 1b), variability of data points (Figure 2d), reliability of CeSnAP (Figure 2a) is added accordingly to the revised manuscript. We hope that inclusion of these data in the revised manuscript will aid other labs who are considering setting up such an assay.

3) One out of five best hits (felodipine), showed to be a false positive, which is something expected. However false negatives are extremely dangerous. Indeed, among the negative ones some have been showed to be neuroprotective in PD models (e.g. prednisone and valproic acid). Moreover, valproate is used with ethosuximide for epilepsy and convulsion treatment. To test for a “false negatives rate” I suggest a post-screen validation of a sampling of at least 5 negative drugs, among those known to be protecting from neurodegeneration in PD models.

We have carried out a post-screen validation of three negative drugs (Supplementary Figure 3d). Riluzole was at 72%, Linagliptin was at 75%, and Milnacipran was at 101% of vehicle-level curling. Therefore, this repeated curling analysis of three negative hits is in agreement with our original screen, showing that false negatives are unlikely.

One possibility is that sporadic PD – which was the source of GWAS data where we originally identified decreased *bcat-1* as a possible PD cause (Yao & Kaletsky, et al. *Nature Biotech* 2018) - may result from different types of genetic lesions, and thus may respond to different drugs. This is important, as some PD drugs might make some PD cases worse, depending on the underlying molecular mechanism. We have added this important point to the manuscript – thank you for pointing it out.

4) A control that the 4 best hits are not influencing the RNAi machinery and therefore its effects in *bcat-1* silencing is needed. For example, silence *unc-54* or *mom-2* and check for the expected phenotypes after treating with drugs.

The reviewer’s concern is that the top drug hits may be lowering curling because they are disrupting the RNAi machinery and therefore *bcat-1* expression may become restored. We have now included in the revised manuscript an experiment in which *bcat-1(RNAi)* worms were moved to *dcr-1* RNAi on day 5, in order to intentionally disrupt RNAi machinery (as shown in Dillin et al., 2002). We found that *dcr-1* knockdown does not rescue motor function on day 8 (Supplementary Figure 3c). Therefore, disrupting RNAi machinery is not sufficient to restore motor function, indicating that the positive results with the top drug hits cannot be explained by RNAi disruption.

Minor comments:

5) There is already one paper counting the number of curls in *C. elegans* mutants, in automated fashion (Hoshi and Shingai, 2006). The authors should add a reference to it and briefly comment advantages and disadvantages of their approach compared to this other one.

Thank you. As kindly pointed out, our original text lacked a full discussion of advantages and limitations of our approach compared to existing methods. In addition to adding the mentioned study, we have carefully updated the manuscript to include a detailed discussion of CeSnAP advantages and limitations compared with prior tracking and curling detection tools.

6) It is important to discuss more the two phenotypes: curled and near-curved positions. Are they part of the same defect or two different manifestations that can be separated? How much are they contributing to the overall sum in all conditions (as shown in 1h)?

In the updated manuscript, we have now included three curling postures, namely Coiled, Curled, and Near-curved (Figure 2b-c). Following the reviewer's suggestion, we carefully investigated how much they are each contributing to the overall sum (Figure 2d). Although these three categories are all manifestations of spasm-like 'curling' behavior, the Coiled posture is shown to distinguish control and *bcat-1* the most effectively (based on mean ratio of *bcat-1* to control curling level) for the purposes of high-throughput screening.

7) In panels 1h and 2b testing of 400 different unknown conditions is described. Taking in account comment 1, how long did it take to screen 400 conditions is an important information to be added.

It took almost two years to optimize the screen conditions and code, and screen 400 conditions, distributed among 75 different experiments, for both this paper as well as experiments reported in our companion paper. However, to repeat the screen, now that we have optimized conditions and developed CeSnAP, should only take a few weeks. In order to have this paper stand apart from our companion paper, we have removed all data that is not unique to the current paper. The main screen of 50 FDA-approved drugs was done in the span of less than 5 months. As recommended, the relevant information has been added to the updated manuscript.

8) Some drugs show an interesting enhancing phenotype, up to 140% of control. This is an important information for twofold screenings. The author should discuss this aspect.

As stated in response to Reviewer 2, our aim was to identify potential therapies that may improve PD-related motor symptoms, and therefore drugs that might worsen the phenotype were not further

pursued. However, our results and the CeSnAP method may open up new avenues of investigation for future studies that may wish to identify factors that worsen curling behavior.

9) In Mat. and Meth. a long list of strains is listed, which I don't get when were used. Please correct or explain their use.

Upon removing all content that is relevant only to our companion paper, we have modified the Materials and Methods section to only include strains used in this self-standing study.

10) It is not clear to me how the snapshots were taken and the technical setup used for the automated acquisition with the motorized microscope. Every how many minutes the same well is photographed again? Please explain in order to allow any researcher to reproduce the work.

To provide greater clarity, Figure 1d has been updated to clearly illustrate the automated snapshot acquisition protocol. An extensive description regarding the linear stage setup has also been added to the revised manuscript, as well as a detailed instruction manual for CeSnAP. It takes linear stages ~2s to take one snapshot and move to the next location. During a 60min time period, platforms can take up to 1800 snapshots. The user can distribute this total based on the number of experimental conditions, replicates, and rounds in their experiments. Depending on the total number of wells in each experiment, it would take 30s to 2min for the same well to be photographed again.

11) There is an evident discrepancy between panel 1f and 1g, which I understand are the same experiment performed on different days. Please discuss it.

As correctly stated, the “Videos Automated” panels in Figure 1f and Figure 1g of the original manuscript are the same experiment performed on different days. Figure 1f depicted worms both in Curled and Near-curved positions, while Figure 1g only used worms in the Curled position for analysis. In the updated manuscript, three curling postures (Coiled, Curled, and Near-curved) are used. To avoid any discrepancy, both figures now use the same metric for analysis (Figure 2b-c). These points are discussed in the revised manuscript.

12) The curly phenotype is an age related one, hence the 8 days after adulthood, which make sense being related to PD. Please make it explicit.

Thank you – a description has been added to explicitly discuss the relationship of age to both curling and PD.

13) *The comment to panel 2A (Furthermore, we found that using snapshots to quantify the curled position distinguished between *bcat-1*(RNAi) and control groups more effectively than the near-curved position) is neither clear nor validated with statistics. Please explain and complete. Moreover, I would move it to Figure 1.*

Figure 2d has been added to replace panel 2A of the original manuscript and has been moved after the comparison figures, as suggested. The probability density of % worms in the three postures (Coiled, Curled, and Near-curved) showed Coiled postures distinguish spasm-like 'curling' motor defect of vehicle-treated *bcat-1*(RNAi) worms with greater mean ratios of *bcat-1* to control than that of Near-curved (Figure 2d). We have added a description of this to the updated manuscript.

14) *If panel 1h is going to be maintained it should be moved to Fig. 2.*

The figure panel has been moved to the suggested location.

15) *From Yao et al., 2018 I understand that the reduction of *bcat-1* in *C. elegans* enhances α -synuclein effects and does not by itself promote dopaminergic neurodegeneration, as stated in page 2. Please correct.*

We have revised the manuscript to correctly describe this, as suggested. *Bcat-1* reduction does induce neurodegeneration in cholinergic neurons, independent of α -synuclein, but dopaminergic neurons are well maintained in *C. elegans* until very advanced ages (eventually it does increase neurodegeneration of dopaminergic neurons as well, but at such an old age (> 18d) that screening is difficult for unrelated reasons).

Reviewer 4 (Remarks to the Author):

*This paper by Sohrabi et al is a cute use of technology to speed up an originally manual and painful assay. The paper demonstrates the full use of this technology in a screen that's relevant to Parkinson's. Had the paper existed on its own, I would have supported the publication, but considering the following factors, it seems to me the more appropriate thing to do is to have this information be a part of the preprint (Mor et al). First, the methodology has limited utility – it is detection curling in *C. elegans*; it is very very specific. It is not useful for other organisms, and not useful for detecting other behavioral changes in *C. elegans*. Second, the methodology is largely based on canned (quite standard) pieces, so there is almost no technical merit, even though the application is extremely elegant and obviously useful.*

Third, the exciting results is metformin showing promises, but the preprint has already disclosed the information and in fact did a much deeper dive on that subject.

We appreciate the reviewer's feedback. **To address the second criticism, the methodology part of the paper has been completely replaced with a new machine learning-based platform that is now two times faster than the previously submitted method.** Although our original approach worked well to complete the drug screen, as stated, it was based on quite standard thresholding techniques. In order to meet the state-of-the-art of other approaches, we have now designed a deep learning algorithm to train a convolutional neural network for detection of worms and classification into multiple posture groups. We have also included a detailed discussion of previously developed tools for tracking worms, along with advantages and limitations of our new technique. Our autonomous analysis program which requires no user supervision coupled with our fast snapshot capturing protocol that maintains low data overhead is mainly what distinguishes our tool from others.

As correctly pointed out, the original version of CeSnAP was specifically developed for curling analysis and had limited utility. We have attempted to address this concern with the new version of CeSnAP, which will be useful for quantifying other behavioral changes and morphological analysis in *C. elegans*. Our simple segment-train-quantify workflow enables users to extract mask images from snapshots in order to train their own neural network (Figure 1e). Future experiments can then be scored in a high-throughput manner using their preferred neural network. Additionally, our open-source platform can easily be modified and adapted for other applications such as high-throughput phenotypic analysis of fluorescence-based assays. It should also be noted that the curling motor defect is not limited to *bcat-1(RNAi)* worms, as demonstrated in Figure 3f.

In terms of the relationship between this study and our companion study, we have taken careful steps so that this article stands on its own. All data reported in our companion paper has been removed, and we now report only data that is unique to the current study. It is worth mentioning that metformin was not the only hit candidate identified in our screen. Additionally, the main purpose of this study is to report the new methodology and a proof-of-concept drug screen that would not have been possible without this workflow. We have included more in-depth discussion on the methodology in the revised manuscript, providing a detailed roadmap for others who may wish to use the segment-train-quantify workflow or high-throughput curling assay for their own investigations. With the addition of our machine learning-based platform in the updated manuscript, we believe that this study now has enough technical merit to be published as a separate article from our companion paper.

References

- Broekmans, O. D., Rodgers, J. B., Ryu, W. S., & Stephens, G. J. (2016). Resolving coiled shapes reveals new reorientation behaviors in *C. elegans*. *Elife*, *5*, e17227.
- Dillin, A., Hsu, A., Arantes-Oliveira, N., Lehrer-Graiwer, J., Hsin, H., Fraser, A. G., Kamath, R. S., Ahringer, J., & Kenyon, C. (2002). Rates of behavior and aging specified by mitochondrial function during development. *Science*, *298*(5602), 2398-2401.
- Ghosh, R., & Emmons, S. W. (2008). Episodic swimming behavior in the nematode *C. elegans*. *Journal of Experimental Biology*, *211*(23), 3703-3711.
- Ikeda, Y., Jurica, P., Kimura, H., Takagi, H., Zbigniew, S., Kiyono, K., Arata, Y., & Sako, Y. (2020). *C. elegans* episodic swimming is driven by multifractal kinetics. *bioRxiv*,
- Murphy, C. T., McCarroll, S. A., Bargmann, C. I., Fraser, A., Kamath, R. S., Ahringer, J., Li, H., & Kenyon, C. (2003). Genes that act downstream of DAF-16 to influence the lifespan of *Caenorhabditis elegans*. *Nature*, *424*(6946), 277-283.
- Roussel, N., Sprenger, J., Tappan, S. J., & Glaser, J. R. (2014). (2014). Robust tracking and quantification of *C. elegans* body shape and locomotion through coiling, entanglement, and omega bends. Paper presented at the *Worm*, *3*(4) e982437.

REVIEWERS' COMMENTS:

Reviewer #1 (Remarks to the Author):

In summary my concerns addressed were

- 1) the use of image stills instead of also including temporal cues
- 2) the discretisation of the behavioural motifs
- 3) the image processing itself.

The revised version introduces a machine learning solution to address comment 1) and 3). This was a wise decision and I am impressed by performance of their system. This was indeed a very good idea. Moreover, the generation of the training data, the usage of data augmentation and the non-oversized CNN used indicate a thorough examination of computer vision techniques - well done!

With respect to my second comment the authors also included a mathematical description of the posture.

In short: all my major concerns were addressed. These more advanced methodologies in combination with the general idea of using curling behaviour as an assay for PD in a drug screening assay now results in a convincing piece of work. I therefore recommend this paper for publication.

Reviewer #2 (Remarks to the Author):

I am satisfied that the authors have addressed the concerns outlined in my initial review.

Reviewer #3 (Remarks to the Author):

The paper highly benefitted from the extensive revisions. Is now more convincing, focused, clear and detailed. It presents a very useful approach for the C.elegans community and possibly for other small animal models. It also helped a lot to have it clearly separated from the companion one, which is now published. Given the level of details provided I think any researcher should be able to reproduce their work.